# GLSIM: Detecting Object Hallucinations in LVLMs via Global-Local Similarity

**Seongheon Park**     **Sharon Li**
Department of Computer Sciences
University of Wisconsin-Madison
{seongheon_park, sharonli}@cs.wisc.edu

## Abstract

Object hallucination in large vision-language models presents a significant challenge to their safe deployment in real-world applications. Recent works have proposed object-level hallucination scores to estimate the likelihood of object hallucination; however, these methods typically adopt either a global or local perspective in isolation, which may limit detection reliability. In this paper, we introduce GLSIM, a novel training-free object hallucination detection framework that leverages complementary global and local embedding similarity signals between image and text modalities, enabling more accurate and reliable hallucination detection in diverse scenarios. We comprehensively benchmark existing object hallucination detection methods and demonstrate that GLSIM achieves superior detection performance, outperforming competitive baselines by a significant margin[1].

## 1 Introduction

Large Vision-Language Models (LVLMs) [1, 2, 3, 4, 5, 6, 7, 8] have made striking advances in understanding real-world visual data, enabling systems that can describe images, answer visual questions, and follow multi-modal instructions with fluency and creativity [9, 10]. Yet beneath this surface of impressive capability lies a critical vulnerability—object hallucinations (OH)—where the model generates plausible-sounding mentions of objects that are not present in the image [11]. An example is illustrated in Figure 1, where the LVLM describes a "dining table" in a birthday party scene, even though the image contains no such object. These hallucinations can undermine user trust, and are particularly concerning in high-stakes domains including medical imaging [1], autonomous navigation [12], and accessibility applications [13]. Detecting such hallucinations is thus essential for safe and reliable deployment of LVLMs, and has become an increasingly active area of research [14].

Existing approaches to object hallucination detection often rely on external knowledge sources, such as human-annotated ground truth annotations [11, 15, 16, 17, 18, 19]. Others prompt or fine-tune external large language or vision-language models as judge to detect hallucinations [20, 21, 22, 23, 24, 25]. However, these approaches face practical limitations: ground-truth references are often unavailable in real-world scenarios, and external LLMs are prone to hallucinating themselves, thereby limiting reliability. This highlights the need for a lightweight, model-internal approach that can detect and self-evaluate hallucinations without supervision or auxiliary models.

In this paper, we propose an object-level hallucination scoring function that operates without relying on external sources, leveraging the embedding similarity between image and text modalities within the latent space of LVLMs. We introduce **G**lobal-**L**ocal **Sim**ilarity (**GLSIM**) score, a method that unify two complementary perspectives: a global similarity score, which captures how well an object semantically fits the overall scene, and a local grounding score, which checks whether any specific

---

[1]Code is available at https://github.com/deeplearning-wisc/glsim

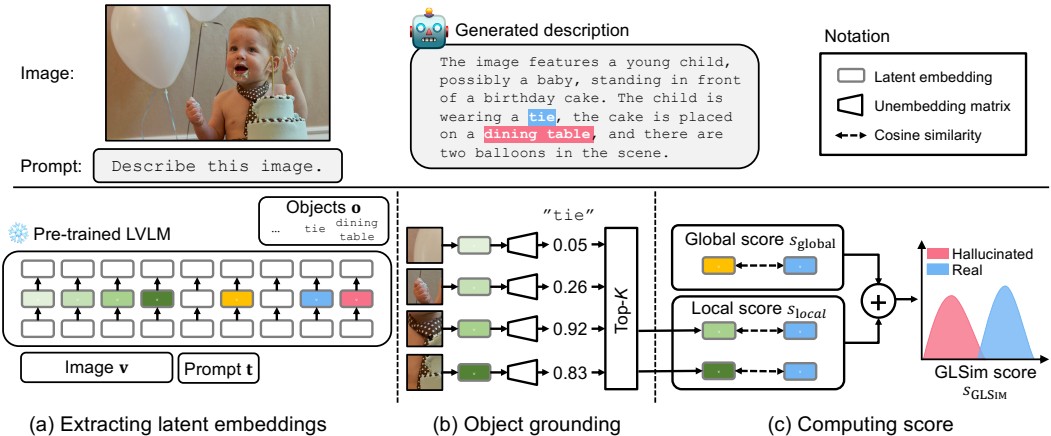

Figure 1: **Overall framework**. (a) We detect object-level hallucinations by leveraging latent embedding similarity. (b) For each object, the most relevant image regions are identified via unembedding from latent image representations. (c) The final GLSIM score is computed as a weighted combination of local (Section 4.2) and global (Section 4.3) signals, capturing both scene-level plausibility and spatial alignment, enhancing object hallucination detection accuracy.

region in the image actually supports the object's presence. This fusion addresses a key shortcoming of prior approaches that rely on one perspective in isolation [26, 27, 28, 29]. For instance, a global-only method may wrongly consider a "dining table" plausible in a birthday party scene (Figure 1), simply because such contextual associations are common in pretraining data—even if no table is visually present. On the other hand, local-only approaches may struggle when a hallucinated object is visually similar to real objects in the scene, as in Figure 2, where a model hallucinates a "handbag" due to confusion with a leather seat of a motorcycle. By integrating both global and local signals, our method can ask not only "does this object belong contextually to the scene?" but also "is there concrete visual evidence for it?", resulting in more accurate, well-rounded, and interpretable hallucination detection across diverse scenarios.

As illustrated in Figure 1, GLSIM works by evaluating each object mention along two axes. First, the global score measures the similarity between the object token's embedding and the overall scene embedding—captured by the final token of the multimodal instruction prompt (highlighted in yellow). Next, we compute a local similarity score that checks for spatial grounding. Specifically, we identify the top image patches most relevant to the object using an adapted Logit Lens technique [30], then assess whether these regions provide strong visual evidence for the object using the average similarity between the object token's embedding and the top-$K$ image token embeddings (highlighted in green). By combining these two complementary signals, GLSIM produces a holistic score that reflects both contextual fit and visual grounding—effectively distinguishing real objects from hallucinations.

We extensively evaluate GLSIM across multiple benchmark datasets and LVLMs, including LLaVA-1.5 [1], MiniGPT-4 [3], and Shikra [31], demonstrating strong generalization and *state-of-the-art* performance in detecting object hallucinations. On both MSCOCO and Objects365 datasets, GLSIM consistently outperforms the latest baselines, including Internal Confidence [28] and attention-based grounding scores [27], achieving up to a **+12.7**% improvement in AUROC. Ablation studies confirm the complementary roles of the global and local components: removing either degrades performance, while their combination yields the most reliable detection. Qualitative results further illustrate how GLSIM accurately flags subtle hallucinations, making it a practical tool for real-world deployment.

Our key contributions are summarized as follows:

1. We propose GLSIM, a novel object hallucination detection method that combines global and local similarity scores between latent embeddings. To the best of our knowledge, this is the first work to demonstrate their complementary effectiveness for the OH detection task.

2. We provide a comprehensive benchmarking of existing OH detection methods, addressing an important gap that has been overlooked in prior work.

3. We demonstrate the superior performance of GLSIM through extensive experiments, conduct in-depth ablations to analyze the contributions of each component and design choice, and verify the generalizability of our method across various LVLMs and datasets.

## 2 Related Works

**Object Hallucination Detection in LVLMs.** Object hallucination (OH) refers to the phenomenon where LVLMs generate textual descriptions that include *non-existent objects* in the image—a critical but underexplored problem in LVLMs with direct implications for reliable decision-making. Such hallucinations can stem from factors including statistical biases in training data [32], strong language model prior [9], or visual information loss [33]. Recent studies have focused on evaluating and detecting OH by leveraging ground-truth annotations [11, 15, 16, 17, 18, 19]. For instance, CHAIR [11] suggests utilizing the discrete ratio of objects presented in the answer relative to a ground-truth object list to identify OH. Another line of work evaluates OH using external LLMs or LVLMs [20, 21, 22, 23, 24, 25]. For instance, GAIVE [22] leverages a stronger LVLM (*e.g.*, GPT-4 [34]) as a teacher to assess the responses of a student model, while HaLEM [35] fine-tunes an LLM (*e.g.*, LLaMA [36]) to score LVLM generations. While effective, these methods are resource-intensive and often lack transparency.

Several recent works have proposed object-level hallucination scores that self-evaluate OH likelihood without requiring an external judge model or additional training. For instance, LURE [26] utilizes the negative log-likelihood (NLL) of the object token generation probability; Internal Confidence (IC) [28] computes the maximum probability of the object token across all image hidden states [30]; and Summed Visual Attention Ratio (SVAR) [27] leverages attention weights assigned to image tokens with respect to the object token. While promising, these methods primarily target hallucination mitigation and often fall short in detection performance: these methods typically leverage either global (*e.g.*, NLL, SVAR) or localized (*e.g.*, IC) signals in isolation and thus fail to capture the nuanced interplay between the overall semantic context and fine-grained visual grounding. Moreover, NLL often fails since LVLMs tend to favor linguistic fluency over factual accuracy [37]; IC does not fully capture contextual information from the generated text; and SVAR can be biased toward previously generated text tokens [38] and vulnerable to attention sink effects [39].

Different from prior works, *we introduce the first object hallucination detection method that explicitly integrates both global and local signals—unifying localized attribution with holistic semantic alignment between the image and generated text*. We benchmark our approach against existing object-level hallucination detection methods across diverse settings to offer a comprehensive comparison in this space. Further related works are provided in Appendix C.2.

## 3 Problem Setup

**Large Vision-Language Models** for text generation typically consist of three main components: a vision encoder (*e.g.*, CLIP [40]) which extracts visual features, a multi-modal connector (*e.g.*, MLP) that projects these visual features into the language space, and an autoregressive language model that generates text conditioned on the projected visual and prompt embeddings.

Given an input image, the vision encoder processes it into a set of patch-level visual embeddings, commonly referred to as visual tokens. These tokens are then projected into the language model's embedding space through the multi-modal connector, resulting in a sequence of $N$ visual embeddings: $\mathbf{v} = \{v_1, \ldots, v_N\} \in \mathbb{R}^{N \times d}$, where each $v_i$ corresponds to a transformed visual token of dimension $d$. On the language side, the input text prompt (e.g., "Describe this image in detail.") is tokenized and embedded into a sequence of language embeddings: $\mathbf{t} = \{t_1, \ldots, t_L\} \in \mathbb{R}^{L \times d}$, where $L$ is the prompt length. These two modalities—the projected visual tokens $\mathbf{v}$ and the textual embeddings $\mathbf{t}$—are concatenated and passed as the input sequence to the language model. The language model then generates a sequence of output tokens: $\mathbf{y} = \{y_1, \ldots, y_M\}$, where each $y_i \in \mathcal{V}$ is drawn from a vocabulary space and $M$ is the output length.

**Object hallucination detection.** In this work, we focus on detecting *object existence hallucination* in LVLMs—cases where the model generates text that references objects not present in the image [15, 41, 14]. This represents the most fundamental and critical form of errors affecting model reliability. We provide the formal task definition below.

**Definition 3.1** (**Object Hallucination Detector**). *Let* $\mathbf{x} = (\mathbf{v}, \mathbf{t})$ *denote the input to the LVLM, and* $\mathbf{y} = \{y_1, \ldots, y_M\}$ *be the sequence of generated tokens from the model. From* $\mathbf{y}$*, we extract a set of object mentions* $\mathbf{o} = \{o_1, \ldots, o_{n_h+n_r}\} \subset \mathcal{O}$*, where* $n_h$ *and* $n_r$ *denote the number of hallucinated and real objects, respectively. The task of object hallucination detection is to design a scoring*

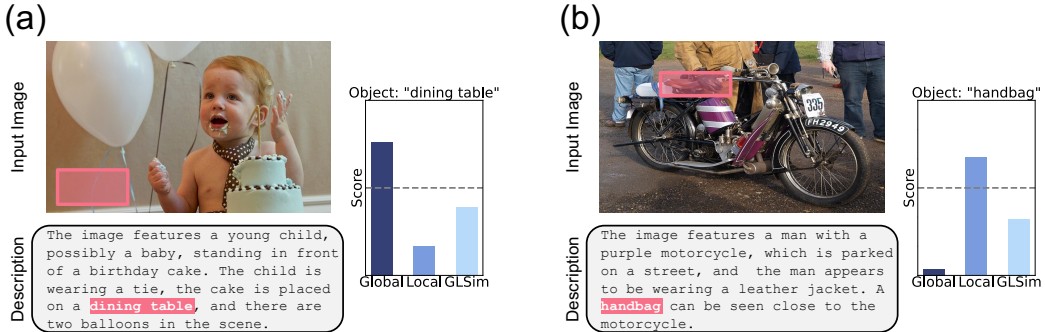

Figure 2: **Qualitative evidence.** In the generated descriptions, hallucinated objects are highlighted in red. The localized image regions are shaded with the same color as their corresponding objects. The gray line shows a threshold value $\tau$. If an object's score is lower than the threshold $\tau$, we consider it a hallucination. In (a), the local score successfully compensates for the failure of the global score, while in (b), the global score offsets the limitations of the local score.

*function $s : \mathcal{O} \times \mathcal{X} \rightarrow [0, 1]$, where $s(o, \mathbf{x})$ quantifies the likelihood that object $o \in \mathcal{O}$ is present in the input $\mathbf{x} \in \mathcal{X}$. Here $\mathcal{O}$ and $\mathcal{X}$ denote the space of objects and input, respectively. Based on this score, we define the object hallucination detector:*

$$G(o, \mathbf{x}) = \begin{cases} 1, & \text{if } s(o, \mathbf{x}) \geq \tau \\ 0, & \text{otherwise}, \end{cases} \tag{1}$$

*where $\tau \in [0, 1]$ is a decision threshold. Here, $G(o, \mathbf{x}) = 1$ indicates that object $o$ is real (i.e., occurs in the image), while $G(o, \mathbf{x}) = 0$ indicates a hallucinated object.*

## 4 Method

**Overview.** In this section, we propose an object-level hallucination scoring function that operates without relying on external sources, leveraging the embedding similarity between image and text modalities within the latent space of LVLMs. We introduce **G**lobal-**L**ocal **Sim**ilarity (GLSIM) score, a method that leverages both global and local similarity measures, and discuss how these complementary signals contribute to effective object hallucination detection.

### 4.1 Motivation: Both Local and Global Signals Matter

Object hallucination in LVLMs often arises when models generate plausible-sounding descriptions that are not visually grounded. But detecting such hallucinations is challenging: they can stem from subtle biases, background patterns, or statistical co-occurrence in training data [15, 42, 26]. Critically, relying on a single perspective—either a global similarity or a local region-level score—is often not enough to reliably catch them. In particular, global similarity quantifies *how semantically related the object is to the image as a whole*. It captures holistic alignment between the object mention and the overall scene, and is useful for assessing whether the object "makes sense" in context. In contrast, local similarity measures *how well the object is visually grounded in a specific region*. It focuses on fine-grained evidence aligned with spatial areas most relevant to the object, helping verify whether it is actually present.

**Qualitative evidence.** Figure 2 illustrates how each signal alone can be insufficient. In panel (a), the LVLM-generated description includes a "dining table", yet no table is present in the image. A global similarity score fails to flag this hallucination—likely because the overall scene (*e.g.*, birthday cake, party setting) frequently co-occurs with tables in training data, leading to a high false-positive signal. In contrast, a local score that focuses on the visual region associated with "dining table" correctly assigns a low similarity, reflecting the absence of meaningful grounding in that region. In contrast, panel (b) shows a failure case for local similarity. The model hallucinates a "handbag," and while the global similarity correctly captures that the handbag is not semantically compatible with the overall scene, the local score becomes unreliable—likely due to a visually similar object in the image (*i.e.*, leather seat of the motorcycle).

These examples underscore the inherent limitations of using either signal in isolation. Global similarity can be overly influenced by high-level contextual associations, leading to false positives when hallucinated objects are contextually plausible within the scene but not visually present. On the other hand, local similarity is sensitive to spatial precision, but can misfire when localization is noisy or there are visually similar objects. As a result, each signal captures only a partial view of the grounding problem. To overcome this, we propose a unified approach, **G**lobal-**L**ocal **Sim**ilarity (GLSIM), that leverages the complementary strengths of both perspectives and offers more accurate and reliable detection of object hallucinations across a diverse range of visual-textual contexts. In the next subsections, we introduce the score definition in detail—explaining how we design global and local similarity for each object mention, and how they are integrated into a single decision score for hallucination detection. More qualitative results are presented in Appendix B.1.

## 4.2 Object Grounding via Local Similarity

A key component of our approach is the computation of the local similarity score, which captures how well an object mention is visually grounded in a specific region of the image. Unlike global similarity, which reflects scene-level plausibility, the local score focuses on verifying the presence of the object at the spatial level. The main challenge lies in identifying the most relevant region for each object mention—without relying on external annotations or bounding boxes.

**Unsupervised object grounding.** We leverage an unsupervised approach that leverages internal representations of the LVLM itself, to ground whether a predicted object token $o$ is hallucinated or not. Given the LVLM input $\mathbf{x} = (\mathbf{v}, \mathbf{t})$, where $\mathbf{v} = \{v_1, \ldots, v_N\}$ are the visual tokens and $\mathbf{t}$ are the prompt embeddings, we extract the hidden representations $h_l(v_i) \in \mathbb{R}^d$ of each visual token $v_i$ at decoder layer $l$. To project these representations into the vocabulary space, we can leverage Visual Logit Lens (VLL) as:

$$\text{VLL}_l(v_i) = h_l(v_i) \cdot W_U,$$

where $W_U \in \mathbb{R}^{d \times |\mathcal{V}|}$ is the unembedding layer matrix. Unlike the original Logit Lens [30], which operates solely in language models, our approach adapts it to a multimodal setting to attribute generated object mentions to relevant visual tokens. We apply a softmax and extract the predicted probability for the target object token $o$: $\text{softmax}(\text{VLL}_l(v_i))[o]$, probability quantifies how likely a visual token $v_i$ is to predict the object word $o$, offering a model-internal signal of relevance between the image patch and object token. Importantly, we select the Top-$K$ image patches with the highest probabilities as the localized regions corresponding to the object $o$:

$$\mathcal{I}(o) = \text{TopK}_{v_i \in \mathbf{v}} \left( \{\text{softmax}(\text{VLL}_l(v_i))[o]\} \right). \tag{2}$$

We visualize object grounding results in Section 5.3 and Appendix B.2.

**Local similarity score.** Based on the localized regions $\mathcal{I}(o)$, we compute average cosine similarity between each localized image embedding and object embedding:

$$s_{\text{local}}(o, \mathbf{x}) = \frac{1}{K} \sum_{v_i \in \mathcal{I}(o)} \text{sim}(h_l(v_i), h_{l'}(o)), \tag{3}$$

where $\text{sim}(\cdot, \cdot)$ denotes cosine similarity, and $l'$ is the decoder layer used to represent the text embedding at the position of the object word. The score should be higher for real objects and relatively lower for hallucinated objects.

## 4.3 Scene-Level Grounding via Global Similarity

While the local similarity score focuses on spatially grounding an object in specific image regions, it alone may be insufficient—especially in cases where localization is ambiguous. To complement this, we introduce a *global similarity score* that measures scene-level semantic coherence between an object mention and the entire image. This can be useful for identifying out-of-context hallucinations (*e.g.*, referencing a "handbag" in a motorcycle scene).

**Global similarity score.** We compute the global similarity as the cosine similarity between the embedding of the object/text token and the embedding of the final token in the instruction prompt. The final instruction token often encodes a condensed summary of the model's understanding of both

image and prompt context. By comparing the object token to this representation, the global score quantifies how well the object semantically aligns with the overall scene. This allows the model to down-weight mentions that may be contextually implausible, even if they are locally aligned with some visual region.

Formally, given an object mention $o$ and LVLM input $\mathbf{x} = (\mathbf{v}, \mathbf{t})$, let $h_{l'}(o) \in \mathbb{R}^d$ be the object token representation at layer $l'$, and let $h_l(\mathbf{v}, \mathbf{t}) \in \mathbb{R}^d$ be the hidden representation of the last visual-text prompt token at layer $l$. The global similarity score is then defined as:

$$s_{\text{global}}(o, \mathbf{x}) = \text{sim}\left(h_l(\mathbf{v}, \mathbf{t}), h_{l'}(o)\right), \tag{4}$$

where $\text{sim}(\cdot, \cdot)$ denotes cosine similarity.

**Global-Local Similarity (GLSIM) score.** To fully leverage the complementary strengths of both grounding signals, we define the final hallucination detection score as a weighted combination of local and global similarity. Specifically, we define the GLSIM score as:

$$s_{\text{GLSIM}}(o, \mathbf{x}) = w \cdot s_{\text{global}}(o, \mathbf{x}) + (1 - w) \cdot s_{\text{local}}(o, \mathbf{x}), \tag{5}$$

where $w \in [0, 1]$ is a hyperparameter controlling the balance between local evidence and global context. This fused score captures both spatial alignment and scene-level plausibility, enabling more accurate detection of hallucinated objects. In practice, we find that a moderate value of $w$ (e.g., 0.6) yields consistently strong performance across diverse scenarios (see Section 5.3). Based on the scoring function, the object hallucination detector is $G(o, \mathbf{x}) = \mathbb{1}\{s_{\text{GLSIM}}(o, \mathbf{x}) \geq \tau\}$, where 1 indicates a real object and 0 indicates a hallucinated object.

## 5  Experiments

### 5.1  Setup

**Datasets and models.** We utilize the MSCOCO dataset [43], which is widely adopted as the primary evaluation benchmark in numerous LVLM object hallucination studies and contains 80 object classes. In addition, we employ the Objects365 dataset [44], which offers a more diverse set of images and a larger category set comprising 365 object classes, along with denser object annotations per image. For evaluation, we randomly sample 5,000 images each from the validation sets of MSCOCO and Objects365. We conduct experiments on three representative LVLMs: LLaVA-1.5 [1], MiniGPT-4 [3], and Shikra [31]. For LLaVA-1.5, we evaluate both 7B and 13B model variants to study scalability. Implementation details are provided in Appendix A. We evaluate on three additional LVLMs—InstructBLIP [2], LLaVA-NeXT-7B [45], Cambrian-1-8B [46], Qwen2.5-VL-7B [47], and InternVL3-8B [48] in Appendix D.1.

**Evaluation.** We formulate the object hallucination detection problem as an object-level binary classification task, where a positive sample is a real object and a negative sample is a hallucinated object. We extract objects from the generated descriptions and perform exact string matching against the ground-truth object classes of each image and their synonyms, following CHAIR [11]. To evaluate OH detection performance, we report: (1) the area under the receiver operating characteristic curve (AUROC), and (2) the area under the precision-recall curve (AUPR), both of which are threshold-independent metrics widely used for binary classification tasks.

**Baselines.** We compare our approach against a comprehensive set of baselines, categorized as follows: (1) *Token probability*-based approaches—Negative Log-Likelihood (NLL) [26] and Entropy [49]; (2) *Logit Lens probability*-based approach—Internal Confidence [28]; (3) *Attention*-based approach—Summed Visual Attention Ratio (SVAR) [27]; and (4) *Embedding similarity*-based approach—Contextual Lens♠ [29]. To ensure a fair comparison, we evaluate all baselines on identical test sets using the default experimental configurations provided in their respective papers. As Contextual Lens♠ was originally proposed for sentence-level hallucination detection, we adapt it for object-level hallucination detection. Further details of these baselines are discussed in Appendix C.

### 5.2  Main results

As shown in Table 1, we compare our method, GLSIM, with competitive object hallucination detection methods, including the latest ones published in 2025. GLSIM consistently outperforms existing

| Dataset | Method | | LLaVA-1.5-7B | | LLaVA-1.5-13B | | MiniGPT-4 | | Shikra | |
|---|---|---|---|---|---|---|---|---|---|---|
| | | | AUROC ↑ | AUPR ↑ | AUROC ↑ | AUPR ↑ | AUROC ↑ | AUPR ↑ | AUROC ↑ | AUPR ↑ |
| MSCOCO | NLL [26] | ICLR'24 | 63.7 | 84.9 | 63.1 | 86.1 | 59.4 | 81.2 | 60.4 | 82.1 |
| | Entropy [49] | ICLR'21 | 64.0 | 85.0 | 63.2 | 86.3 | 60.6 | 83.2 | 62.9 | 84.0 |
| | Internal Conf. [28] | ICLR'25 | 72.9 | 89.3 | 71.0 | 90.0 | 75.7 | 93.0 | 69.1 | 88.5 |
| | SVAR [27] | CVPR'25 | 74.7 | 91.2 | 75.2 | 92.9 | 83.6 | 95.9 | 70.7 | 89.1 |
| | Contextual Lens♠ [29] | NACCL'25 | 75.4 | 90.7 | 78.7 | 92.8 | 84.9 | 96.2 | 69.5 | 87.6 |
| | **GLSIM (Ours)** | | $83.7^{\pm0.3}$ | $94.2^{\pm0.2}$ | $84.8^{\pm0.5}$ | $95.8^{\pm0.2}$ | $87.0^{\pm0.4}$ | $97.0^{\pm0.1}$ | $83.0^{\pm0.7}$ | $94.9^{\pm0.3}$ |
| Objects365 | NLL [26] | ICLR'24 | 62.9 | 60.8 | 59.4 | 61.0 | 56.7 | 70.4 | 58.9 | 64.8 |
| | Entropy [49] | ICLR'21 | 63.3 | 60.9 | 59.1 | 60.4 | 57.3 | 70.7 | 60.7 | 67.6 |
| | Internal Conf. [28] | ICLR'25 | 68.7 | 67.4 | 65.5 | 70.0 | 68.5 | 75.0 | 64.4 | 72.5 |
| | SVAR [27] | CVPR'25 | 64.9 | 66.6 | 63.5 | 68.2 | 71.0 | 79.4 | 60.6 | 68.3 |
| | Contextual Lens♠ [29] | NACCL'25 | 63.2 | 62.6 | 62.1 | 65.6 | 70.2 | 77.8 | 59.6 | 67.0 |
| | **GLSIM (Ours)** | | $72.6^{\pm0.5}$ | $74.6^{\pm0.4}$ | $70.4^{\pm0.8}$ | $74.0^{\pm0.6}$ | $74.8^{\pm0.6}$ | $82.4^{\pm0.7}$ | $69.7^{\pm1.0}$ | $75.9^{\pm0.9}$ |

Table 1: **Main results**. Comparison with competitive object hallucination detection methods on different datasets. For our method, the mean and standard deviation are computed across three different random seeds. All values are percentages, and the best results are shown in **bold**.

state-of-the-art approaches across different models and datasets by a significant margin. Specifically, on the MSCOCO dataset with LLaVA-1.5-7B, GLSIM outperforms SVAR by **9.0%** AUROC, and achieves an **8.3%** AUROC improvement over Contextual Lens, an embedding similarity-based baseline. Unlike Contextual Lens, which relies on the maximum cosine similarity between text embeddings and all image embeddings, GLSIM integrates global and local signals, resulting in more robust detection performance. Notably, our method also demonstrates strong performance on Shikra, achieving a **12.7%** improvement in AUROC on the MSCOCO dataset compared to SVAR. Given that Shikra is trained with a focus on region-level inputs and understanding, this result suggests that our method is effective in models with strong spatial alignment capabilities.

**Comparison with Internal Confidence.** Recently, the Internal Confidence (IC) method [28] was proposed to detect hallucinations using visual logit lens probabilities. Our approach differs from IC in three key ways. First, IC directly uses the maximum probability from the visual logit lens across all image patches and layers, which can be overconfident—assigning high scores to hallucinated objects (see Figure 3). In contrast, we compute the semantic *similarity in representation space* between the object token embedding and the Top-$K$ visual tokens, yielding a more reliable and semantically meaningful signal. For hallucinated objects, this leads to alignment with semantically irrelevant regions, resulting in lower similarity scores, thereby enabling more reliable object hallucination detection. Second, IC considers only the most probable patch, while we aggregate over the Top-$K$ most relevant patches. As shown in our ablation study (Section 5.3), using multiple visual regions improves performance by capturing spatially distributed evidence and reducing sensitivity to local noise. Third, IC is purely local in nature, whereas our framework also integrates a global similarity score that captures object-scene coherence at the image level. Together, these advantages enable our method to outperform IC by a substantial margin of **10.8%** AUROC.

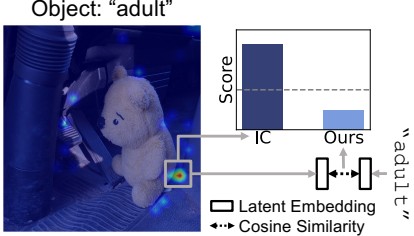

Figure 3: Internal Confidence (IC) can assign high confidence to incorrect regions for hallucinated objects. Our local score (Section 4.2) mitigates this by cross-modal embedding similarity.

## 5.3 Ablation Studies

In this section, we provide various in-depth analysis of each component of our method. All experiments are conducted using LLaVA-1.5-7B and Shikra on the MSCOCO dataset, and results are reported in terms of AUROC (%). Further ablation studies are provided in Appendix D.

**Analysis of global and local scores.** We systematically compare global and local scores on the MSCOCO dataset, as shown in Table 2. **Finding 1: Embedding similarity is an effective scoring metric.** Embedding similarity (ES)-based methods consistently outperform other scoring functions, with GLSim (Top-$K$) achieving a 22.6% improvement over NLL on Shikra. In contrast, token probability (TP)-based approaches are optimized for linguistic fluency rather than object existence accuracy; attention weight (AT)-based methods often fail to align with causal attributions [50]; and

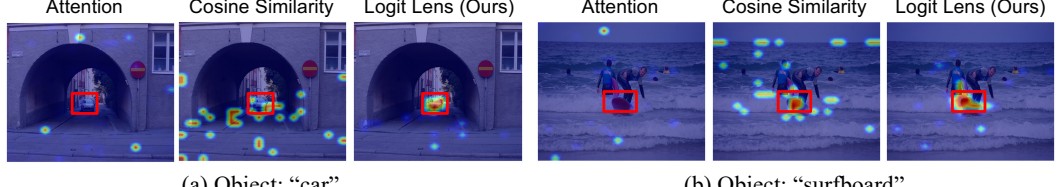

| Attention | Cosine Similarity | Logit Lens (Ours) | | Attention | Cosine Similarity | Logit Lens (Ours) |

(a) Object: "car"  (b) Object: "surfboard"

Figure 4: Object grounding results with LLaVA. Ground-truth bounding boxes are shown in red.

| | Method | Metric | LLaVA | Shikra |
|---|---|---|---|---|
| Global | NLL [26] | TP | 63.7 | 60.4 |
| | Entropy [49] | TP | 64.0 | 62.9 |
| | SVAR [27] | AT | 74.7 | 70.7 |
| | $s_{\text{global}}$ (Eq. (4)) | ES | 79.3 | 78.9 |
| Local | Internal Conf. [26] | LLP | 72.9 | 69.1 |
| | Contextual Lens♠ [27] | ES | 75.4 | 69.5 |
| | $s_{\text{local}}$ (Top-1) | ES | 76.5 | 73.1 |
| | $s_{\text{local}}$ (Top-$K$) | ES | 78.8 | 76.8 |
| G & L | $s_{\text{GLSIM}}$ (Top-1) | ES | 82.0 | 81.0 |
| | $s_{\text{GLSIM}}$ (Top-$K$) | ES | **83.7** | **83.0** |

Table 2: Comparison of global and local scores.

| Score | Grd. Method | LLaVA | Shikra |
|---|---|---|---|
| $s_{\text{global}}$ | - | 79.3 | 79.8 |
| $s_{\text{local}}$ | Attention | 66.3 | 65.0 |
| | Cosine Sim. | 76.2 | 70.1 |
| | Logit Lens | 78.8 | 76.8 |
| $s_{\text{GLSIM}}$ | Attention | 79.4 | 80.0 |
| | Cosine Sim. | 80.7 | 80.9 |
| | Logit Lens | 83.7 | 82.0 |

Table 3: Object grounding methods.

Logit Lens probability (LLP) methods tend to exhibit overconfidence. By directly capturing the semantic alignment between image and text modalities, embedding similarity provides a more reliable signal for OH detection. **Finding 2: Object grounding improves OH detection.** Among local methods, our approach leverages grounded objects in the image and computes embedding similarity directly with those object representations, achieving a 7.7% improvement over Internal Confidence on the Shikra model. This enables fine-grained alignment, unlike Internal Confidence and Contextual Lens methods, which rely only on the maximum token probability or cosine similarity score. **Finding 3: Combining global and local scores further improves performance.** By combining global ($s_{\text{global}}$) and local ($s_{\text{local}}$) similarity scores, we observe additional gains of 2.7% in Top-1 and 4.4% in Top-$K$ for the LLaVA model. This demonstrates that our scoring function design in Equation (5) effectively integrates the complementary strengths of both global and local signals.

**Comparison of object grounding methods.** We explore several design choices for the patch selection for object grounding in Section 4.2, with results summarized in Table 3. Specifically, we vary the metric used for Top-$K$ ($K = 32$) patch selection, comparing (1) attention weights, (2) cosine similarity, and (3) our method (visual logit lens). Our method outperforms attention weights by 12.5% and cosine similarity by 2.6% in local score evaluation. When combining global and local scores, our method achieves gains of 4.3% over attention weights and 3.0% over cosine similarity. We further visualize the Top-$K$ patch scores for each metric in Figure 4. From the visualization, we observe that high attention weights tend to be assigned to irrelevant regions [39]; cosine similarity better localizes object regions but still assigns spuriously high scores to background areas. In contrast, ours accurately highlights object regions, leading to more reliable patch selection for grounding.

**Design choices for global and local scores.** We ablate several key design choices for each scoring function in Table 4. For the global score ($s_{\text{global}}$), we compare (1) similarity with the last image token embedding, (2) average similarity across all image tokens, and (3) similarity with the last instruction token. The last instruction token performs best, outperforming the average similarity by $8\%$, highlighting its strength in capturing scene-level semantics. For the local score ($s_{\text{local}}$), we compare (1) a Logit Lens probability-weighted average of local similarities among top-$K$ patches and (2) a non-weighted average as in Equation (3), where the latter works slightly better. Finally, combining global and local scores improves performance for

| Score | Method | LLaVA | Shikra |
|---|---|---|---|
| Global | Last image token | 65.9 | 56.2 |
| | Average image token | 71.3 | 66.7 |
| | Last instruction token | 79.3 | 79.8 |
| Local | Weighted average | 75.8 | 73.0 |
| | Non-weighted average | 78.8 | 76.8 |
| GLSIM | Average & Eq. (3) | 79.2 | 77.0 |
| | Last inst. & Eq. (3) | 83.7 | 82.0 |

Table 4: Design choices for global and local scoring functions.

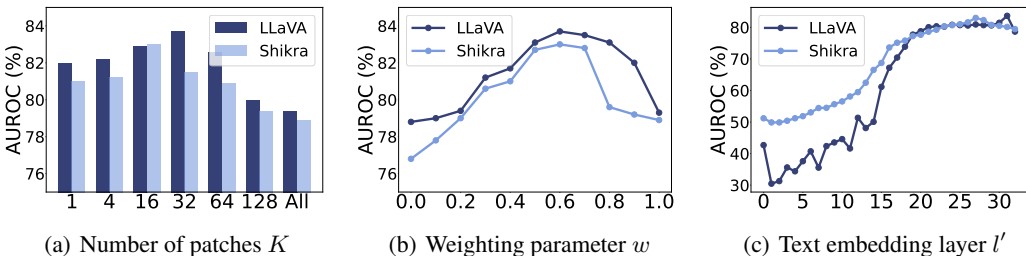

(a) Number of patches $K$      (b) Weighting parameter $w$      (c) Text embedding layer $l'$

Figure 5: (a) Effect of the number of selected image patches $K$; (b) effect of the weighting parameter $w$ in Equation (5); and (c) effect of the text embedding layer index $l'$.

both variants of the global score. This confirms that the two signals are complementary and supports the design of our scoring function.

**How do the selected number of patches $K$ affect the performance?** We analyze the impact of varying the number of selected patches $K$ in Equation (2) on object hallucination detection performance in Figure 5(a). Performance improves with increasing $K$ up to $K$=32 for LLaVA and $K$=16 for Shikra, after which it degrades. This trend suggests that a small $K$ may fail to capture sufficient object information, while a large $K$ introduces irrelevant regions, adding noise. Given that LLaVA processes 576 image tokens, the optimal $K$ roughly corresponds to 6% of total image tokens. This highlights the importance of choosing $K$ relative to input resolution for effective OH detection.

**How does the weighting parameter $w$ affect performance?** In Figure 5(b), we examine the effect of the weighting parameter $w$ in Equation (5) on OH detection performance. Performance increases with $w$ up to 0.6, after which it declines. Smaller values of $w$ place greater emphasis on the local score, while larger values prioritize the global score. We find that moderate values consistently yield the best results across models, suggesting that global and local signals are complementarily informative—where the global score captures scene-level semantics, and the local score captures fine-grained, spatial-level semantics. These results support our design choice of combining both components through a balanced weighting scheme, effectively enhancing overall performance.

**How does the text embedding layer index affect performance?** We examine how the choice of text embedding layer $l'$ influences overall performance when computing embedding similarity in Equation (3) and Equation (4). We fix the image embedding layer $l$ to the 32nd layer for LLaVA and the 30th layer for Shikra, as specified in Table 5. As shown in Figure 5(c), the best performance is achieved at the 31st layer for LLaVA and the 27th layer for Shikra. Performance improves with later layers, suggesting that semantic representations are progressively refined in later layers. However, it slightly drops afterward, which supports the observation from [51] that the optimal layer for downstream tasks may not necessarily be the final layer. These findings indicate that later-intermediate layers are particularly effective for object hallucination detection. The complete performance matrix over all $(l, l')$ layer pairs is provided in Appendix E.

## 6 Conclusion

In this paper, we propose GLSIM, a novel training-free framework for object hallucination detection, which exploits the complementary strengths of global scene-level semantics and fine-grained spatial alignment by leveraging embedding similarity. Empirical results demonstrate that GLSIM achieves superior performance across diverse families of LVLMs and two representative datasets. Our in-depth quantitative and qualitative ablations provide further insights into understanding the effectiveness of GLSIM. We hope our work will inspire future research on OH detection from diverse perspectives.

**Limitations and future work.** Our analysis in this paper focuses on object existence hallucinations, as annotations and benchmarks for attribute and relation hallucinations are currently limited. Nonetheless, it would be interesting to investigate further the grounding ability of the Logit Lens technique for attributes and relationships to quantify local similarity beyond object presence. Moreover, leveraging accurate OH detection from our method to guide model editing or prediction refinement presents a promising future direction for mitigating object hallucinations.

## Acknowledgement

We gratefully acknowledge Changdae Oh and Hyeong Kyu Choi for their valuable comments on the draft. Seongheon Park and Sharon Li are supported in part by the AFOSR Young Investigator Program under award number FA9550-23-1-0184, National Science Foundation under awards IIS-2237037 and IIS2331669, Alfred P. Sloan Fellowship, and Schmidt Sciences Foundation.

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

# Appendix

## Contents

## A Implementation Details

We implement our method using greedy decoding with a maximum generated token length of 512. The layer indices $(l, l')$, the number of selected patches $K$, and the weighting parameter $w$ used for computing the final score are selected based on a separate validation set, as detailed in Table 5. For multi-token objects, we use the first token to compute the scores and consider the first occurrence of each object for hallucination detection. The total number of generated objects is shown in Table 6. For all experiments, we report the average over three different random seeds. All experiments are conducted using Python 3.11.11 and PyTorch 2.6.0 [52], on a single NVIDIA A6000 GPU with 48GB of memory.

| Model | Hyperparameters | | |
|---|---|---|---|
| | Layer indices | $K$ | $w$ |
| LLaVA-1.5-7b | (32, 31) | 32 | 0.6 |
| LLaVA-1.5-13b | (40, 38) | 32 | 0.6 |
| MiniGPT-4 | (32, 30) | 4 | 0.5 |
| Shikra | (30, 27) | 16 | 0.6 |

Table 5: Hyperparameters.

| Model | MSCOCO | | Objects365 | |
|---|---|---|---|---|
| | Real | Hallu. | Real | Hallu. |
| LLaVA-1.5-7b | 14,910 | 4,121 | 14,357 | 9,850 |
| LLaVA-1.5-13b | 15,372 | 3,687 | 15,086 | 9,672 |
| MiniGPT-4 | 11,642 | 2,282 | 11,603 | 7,222 |
| Shikra | 15,724 | 4,727 | 15,063 | 10,350 |

Table 6: Number of generated objects.

# B  Additional Qualitative Results

## B.1  Qualitative Results

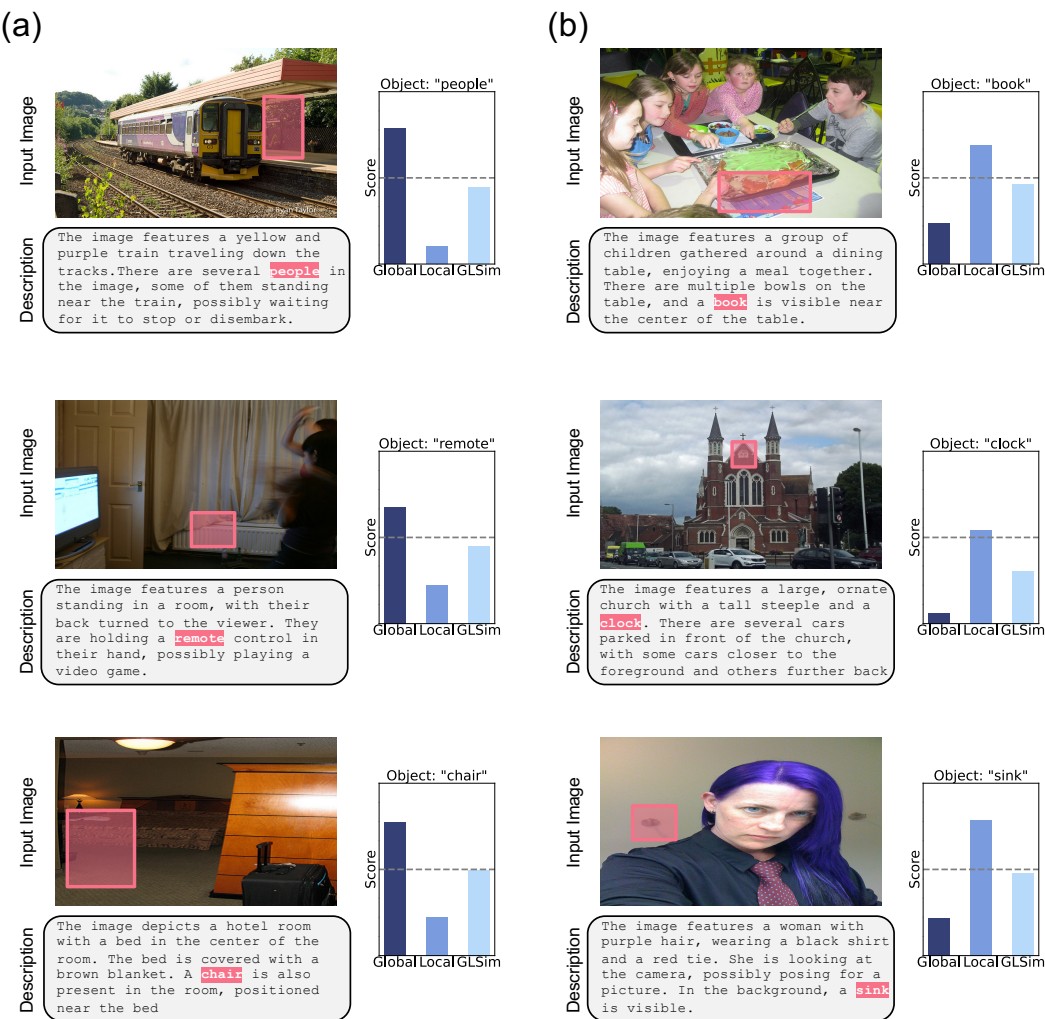

Figure 6: **Additional qualitative evidence.** In the generated descriptions, hallucinated objects are highlighted in red. The localized image regions are shaded with the same color as their corresponding objects. The gray line shows a threshold value $\tau$. If an object's score is lower than the threshold $\tau$, we consider it a hallucination. In (a), the local score successfully compensates for the failure of the global score, while in (b), the global score offsets the limitations of the local score.

## B.2 Object Grounding

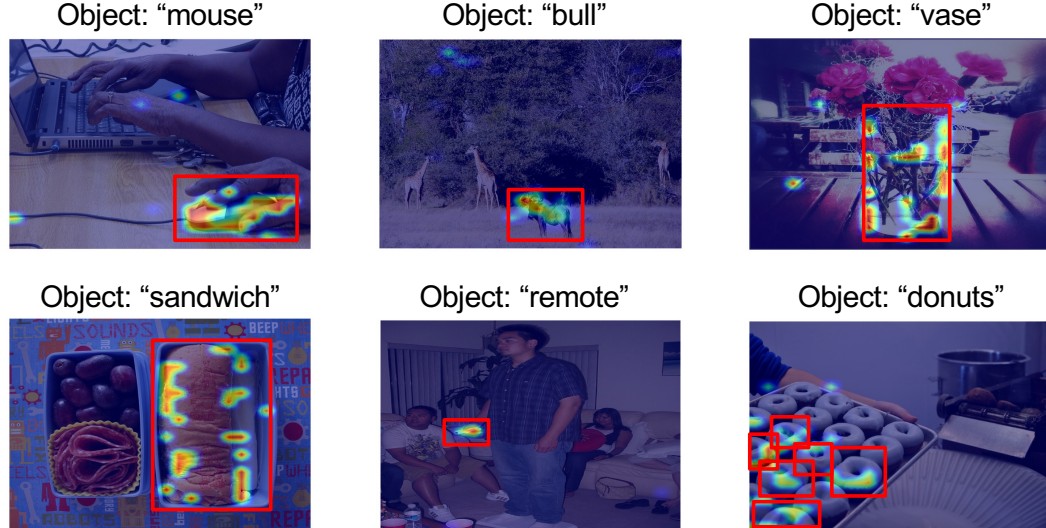

Figure 7: **Object grounding results for real objects.** We visualize the Top-$K$ Logit Lens probabilities at the 32nd layer of LLaVA-1.5-7B. Ground-truth bounding boxes are shown in red.

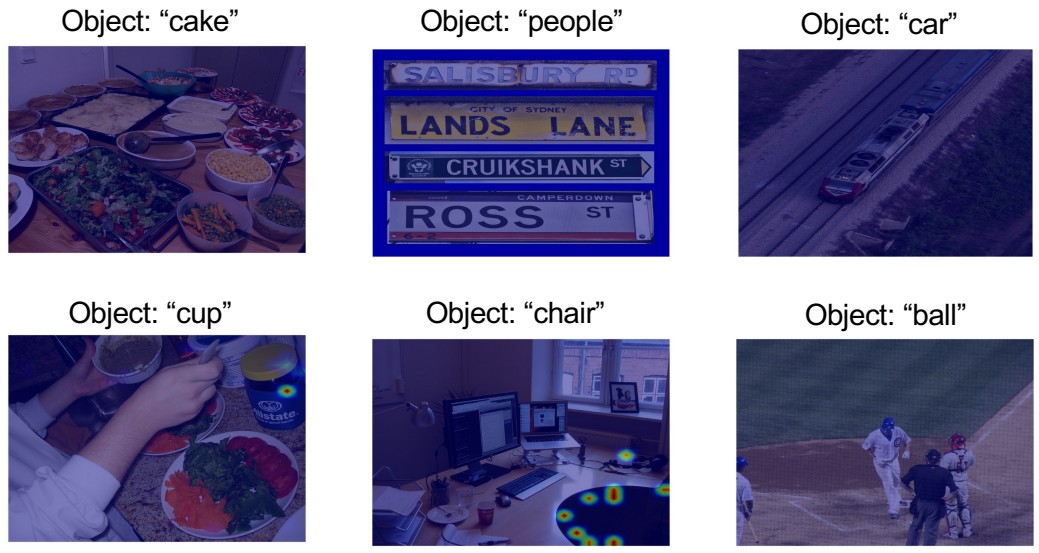

Figure 8: **Object grounding results for hallucinated objects.** We visualize the Top-$K$ Logit Lens probabilities at the 32nd layer of LLaVA-1.5-7B.

## C Related Works

### C.1 Baselines

**Negative Log-likelihood.** Zhou *et al*. [26] represents the probability of autoregressive decoding for each object token as $p(o \mid \mathbf{y}_{<j}, \mathbf{v})$, where $j$ denotes the positional index of object $o$. For each object $o$, the corresponding hallucination score is defined as:

$$s_{\text{nll}} = -\log p(o \mid \mathbf{y}_{<j}, \mathbf{v}).$$  (6)

To align with the definition in Equation (1), we use $s'_{\text{nll}} = -s_{\text{nll}}$.

**Entropy.** We further investigate the object-level hallucination score by estimating the entropy [49] of the token probability distribution at position $j$:

$$s_{\text{entropy}} = -\sum_{y \in \mathcal{V}} p(y \mid \mathbf{y}_{<j}, \mathbf{v}) \log p(y \mid \mathbf{y}_{<j}, \mathbf{v}). \tag{7}$$

To align with the definition in Equation (1), we use $s'_{\text{entropy}} = -s_{\text{entropy}}$.

**Internal Confidence.** Jiang *et al.* [28] apply the logit lens to image representations, enabling the analysis of how visual features are transformed into textual predictions. To quantify the model's confidence for object hallucination detection, the internal confidence score is computed as the maximum softmax probability of the object word $o$ across all image representations and layers. Following the notations introduced in Section 4.2, the hallucination score is defined as:

$$s_{\text{IC}} = \max_{l \in [L]} \max_{i \in [N]} \text{VLL}_l(v_i)[o], \tag{8}$$

where $L$ denotes the total number of layers and $N$ denotes the total number of image patches.

**Summed Visual Attention Ratio (SVAR).** The Visual Attention Ratio (VAR) quantifies the interaction of a generated token $o$ with visual information by summing its attention weights assigned to image tokens in a specific attention head $h$ and layer $\ell$:

$$\text{VAR}^{(\ell,h)}(o) \triangleq \sum_{i=1}^{N} A^{(\ell,h)}(o, v_i), \tag{9}$$

where $A^{(\ell,h)}(o, v_i)$ represents the attention weight from object token $o$ to image token $v_i$ at $h$-th head in $l$-th layer. Building on this, Jiang *et al.* [27] define the Summed Visual Attention Ratio (SVAR), which measures the overall visual attention by averaging VAR scores across all heads and summing over a range of layers. Specifically, for an object token $o$ within layers $\ell_5$ to $\ell_{18}$, SVAR is computed as:

$$s_{\text{SVAR}} = \frac{1}{H} \sum_{\ell=5}^{18} \sum_{h=1}^{H} \text{VAR}^{(\ell,h)}(o), \tag{10}$$

where $H$ denotes the total number of attention heads.

**Contextual Lens.** To detect sentence-level hallucination, Phukan *et al.* [29] compute the maximum cosine similarity between the average embedding of the generated description at a specific layer $l_T$ and each image embedding at layer $l_I$.

$$\text{Sentence-level Score} = \max_{i \in [N]} \text{sim}\left(\frac{1}{M} \sum_{j=1}^{m} h_{l_T}(y_j), h_{l_I}(v_i)\right). \tag{11}$$

To compute the object-level hallucination score, we modify the original score with:

$$s_{\text{CL}} = \max_{i \in [N]} \text{sim}(h_{l_T}(o), h_{l_I}(v_i)). \tag{12}$$

## C.2 Extended Literature Review

**Sentence-level hallucination detection in LLMs and LVLMs** aims to classify an entire generation as either hallucinated or correct, providing a coarse-grained assessment of factuality [53]. A plethora of work addresses sentence-level hallucination detection in large language models (LLMs) by designing uncertainty scoring functions, such as utilizing token generation probabilities [54], prompting LLMs to quantify their confidence [55], and evaluating consistency across multiple responses [56]. Specifically, internal state-based methods leverage latent model embeddings [57], employing techniques such as contrast-consistent search [58], identifying hallucination-related subspaces [59], or reshaping the latent space for hallucination detection [60].

Recently, reference-free sentence-level hallucination detection for large vision-language models (LVLMs) has attracted research attention. Li *et al*. [61] first compare uncertainty quantification methods from LLMs for application to LVLMs. Inspired by [62], VL-Uncertainty [63] estimates uncertainty by measuring prediction variance across semantically equivalent but perturbed prompts.

In contrast to these works, we propose the hallucination scoring function for *object-level hallucination detection* in LVLMs, which provides a fine-grained assessment by localizing hallucinations within generations rather than classifying entire outputs [64]. Our method leverages latent embeddings from both visual and textual modalities and explores intrinsic metrics tailored to LVLMs.

## D   Further Ablation Studies

### D.1   Results for Additional Models

| Method | | InstructBLIP | LLaVA-NeXT | Cambrian-1 | Qwen2.5-VL | InternVL3 |
|---|---|---|---|---|---|---|
| NLL [26] | ICLR'24 | 65.1 | 56.1 | 50.1 | 59.1 | 55.7 |
| Entropy [49] | ICLR'21 | 65.6 | 57.5 | 50.2 | 59.1 | 55.5 |
| Internal Conf. [28] | ICLR'25 | 81.9 | 77.8 | 65.4 | 60.3 | 63.3 |
| SVAR [27] | CVPR'25 | 78.4 | 76.9 | 60.5 | 70.8 | 68.8 |
| Contextual Lens♠ [29] | NACCL'25 | 83.0 | 70.1 | 63.4 | 65.1 | 65.2 |
| **GLSIM (Ours)** | | **85.0** | **81.4** | **79.7** | **76.1** | **73.2** |

Table 7: Performance on additional models evaluated on MSCOCO.

We further evaluate our approach on five additional advanced large vision-language models—InstructBLIP [2], LLaVA-NeXT-7B [45], Cambrian-1-8B [46], Qwen2.5-VL-7B [47], and InternVL3-8B [48]—using the MSCOCO dataset. Our method consistently surpasses baseline approaches, yielding AUROC improvements of 2.0% on InstructBLIP, 4.2% on LLaVA-NeXT, 14.3% on Cambrian-1, 5.3% on Qwen2.5-VL, and 4.4% on InternVL3. These results highlight the robustness of GLSIM across diverse architectures and model scales.

### D.2   Attribute and Relational Hallucinations

| Method | Attribute Hallucination | | | Relational Hallucination | | |
|---|---|---|---|---|---|---|
| | LLaVA-1.5-7B | LLaVA-1.5-13B | Qwen2.5-VL-7B | LLaVA-1.5-7B | LLaVA-1.5-13B | Qwen2.5-VL-7B |
| NLL | 58.62 | 60.50 | 56.89 | 57.06 | 57.35 | 54.90 |
| Entropy | 52.21 | 55.32 | 55.84 | 55.72 | 56.27 | 55.03 |
| Internal Confidence | 74.24 | 73.67 | 70.06 | 69.38 | 68.94 | 62.09 |
| SVAR | 67.03 | 68.62 | 71.09 | 61.20 | 65.83 | 63.01 |
| Contextual Lens | 74.02 | 75.48 | 71.98 | 66.46 | 69.85 | 64.88 |
| **GLSIM (Ours)** | **77.19** | **78.07** | **74.09** | **70.03** | **73.64** | **68.95** |

Table 8: Comparison of different methods on attribute and relational hallucinations.

Given the lack of comprehensive benchmarks for token-level detection of attribute and relational hallucinations in open-ended generation, our study primarily focuses on object existence hallucinations. Nonetheless, addressing these more complex forms remains an important open challenge for real-world deployment. *Attribute hallucination* refers to assigning incorrect properties to objects (e.g., "a red car" when the car is blue), whereas *relational hallucination* arises when relationships between objects are misstated (e.g., "a cat sitting on a table" when it is under the table).

To explore the applicability of GLSIM in these settings, we conducted an extension study. Specifically, we generated captions for each LVLM using 500 randomly selected images from the MSCOCO validation set and employed GPT-4o [65] to produce pseudo ground-truth annotations for both attribute and relational hallucinations. We then computed token-level GLSIM scores and aggregated them by averaging across attribute–object spans (e.g., "a red car") and object–relation spans (e.g., "a cat sitting on a table"). These aggregated scores served as unsupervised estimates of hallucination likelihood. Despite its simplicity and lack of task-specific modifications, GLSIM demonstrated meaningful detection capabilities and consistently outperformed baseline methods across both hallucination types.

### D.3 Visualization of Score Distributions

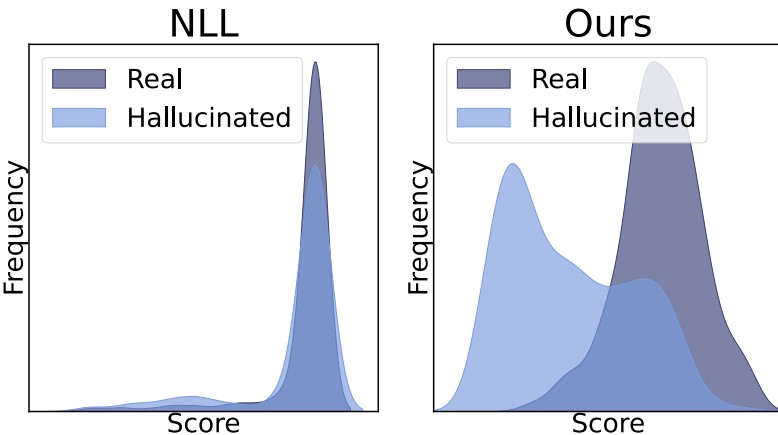

Figure 9: Score distribution for negative log-likelihood [26] *vs.* our method.

We provide score distribution for the negative log-likelihood (NLL) [26] and GLSIM (ours) in Figure 9. Our approach exhibits a more distinct separation between the real and hallucinated data distributions. This enhanced separation highlights the effectiveness of our global and local scoring designs, as well as their combination strategy, contributing to more reliable detection performance compared to existing method.

### D.4 Comparison with POPE

| Model | Method | Time (s) | ACC ↑ | PR (True) ↑ | PR (False) ↑ | Recall ↑ | F1 ↑ |
|---|---|---|---|---|---|---|---|
| LLaVA-1.5-7B | POPE [15] | 8.1 | 79.5 | 80.3 | 70.8 | 96.7 | 87.8 |
| | Ours | 1.6 | 81.8 | 83.1 | 71.8 | 96.6 | 89.3 |
| LLaVA-1.5-13B | POPE [15] | 9.3 | 80.9 | 81.2 | 75.0 | 98.6 | 89.1 |
| | Ours | 1.8 | 83.6 | 86.4 | 75.2 | 95.0 | 90.5 |

Table 9: Comparison with POPE on MSCOCO.

To compare with prompting-based methods such as POPE [15], we extract objects from the generated captions produced using the prompt "Describe this image in detail." We then convert these objects into a set of Yes-or-No short-form questions. We prompt the LVLM with the following template:

> **Input prompt for POPE evaluation**
>
> **Prompt:**
> Q: Is there a {object} in the image?
> A:

Responses containing "Yes" are labeled as 1 (real), and all others as 0 (hallucinated). For evaluation, each question is labeled using the ground-truth object annotations from the MSCOCO dataset, following the same procedure as our main pipeline. We select the decision threshold $\tau$ that maximizes the F1 score. We report accuracy (ACC), precision (PR) for real (true) and hallucinated (false) objects, recall, and F1 score with % in Table 9. To compare computational efficiency, we also report the average inference time (Time) required to detect object hallucinations per image. For LLaVA-1.5-7B, our method achieves 2.8% improvement in the precision of real objects and a substantial 1.0% improvement in the precision of hallucinated objects compared to POPE. From a computational perspective, our method requires only a *single* forward pass for generation, whereas prompting-based methods like POPE require $(1 + C)$ forward passes—one for generating the description and $C$ for the number of object-level verification prompts. Notably, our method reduces inference time by 6.5

seconds per generated description compared to POPE, demonstrating substantial efficiency gains. These results demonstrate the superior effectiveness and computational efficiency of our method in detecting object hallucinations, even when compared to prompting-based approach, particularly in accurately filtering hallucinated content while maintaining precision on real objects.

## D.5 Comparison with External Model-Based Methods

| Method | Time (s) ↓ | Accuracy ↑ | Precision (Real) ↑ | Precision (Halluc.) ↑ | Recall ↑ | F1 ↑ |
|---|---|---|---|---|---|---|
| External-based | 9.3 | 78.6 | 78.9 | 70.8 | **98.5** | 87.6 |
| GLSIM (Internal) | **1.6** | **81.8** | **83.1** | **71.8** | 96.6 | **89.3** |

Table 10: Comparison with external model-based method.

Recent approaches have explored hallucination detection using external knowledge sources, such as large language models (LLMs) and large vision–language models (LVLMs). These methods prompt an external model with a triplet input consisting of the image, instruction, and the generated caption. Such approaches are inherently limited to post-generation evaluation, as hallucinations can only be assessed after the entire caption has been produced. Moreover, they often require multiple forward passes—one to generate the caption from the base model and additional passes for external evaluation—leading to substantial computational overhead.

In contrast, GLSIM operates during the token decoding phase of the base LVLM, enabling real-time hallucination detection at the token level. This property makes GLSIM particularly suitable for interactive or streaming applications where immediate feedback is essential. Importantly, it requires only a single forward pass and does not rely on any external models, thereby avoiding additional uncertainty introduced by potentially hallucination-prone external evaluators.

Table 10 presents a comparison on the MSCOCO dataset using LLaVA-1.5-7B as the base model, and LLaVA-1.5-13B as the external evaluator. GLSIM achieves a substantial inference efficiency advantage, requiring only 1.6 seconds per image compared to 9.3 seconds for the external model-based approach, corresponding to an $82.8\%$ speedup. Despite this efficiency gain, GLSIM also delivers competitive or superior reliability, outperforming the external method across most evaluation metrics. These findings highlight that GLSIM offers a practical and efficient alternative for real-world deployment, combining speed, reliability, and independence from external models in a fully self-evaluating, unsupervised, training-free manner.

## D.6 Multi-token Objects

| Token | LLaVA-1.5-7B | LLaVA-1.5-13B |
|---|---|---|
| First | 83.7 | 84.8 |
| Last | 83.3 | 84.0 |
| Average | 83.4 | 84.2 |

Table 11: Comparison of token selection strategies for multi-token objects.

To compute the visual logit lens probability for object grounding and the embedding similarity for hallucination detection, we default to using the first token of multi-token objects. To evaluate the impact of this design choice, we conduct an ablation study comparing three strategies: (1) using the first token, (2) using the last token, and (3) taking the average across all tokens. Results on the MSCOCO dataset in Table 11 show that the first-token strategy is most effective, since the first token often captures the core semantic meaning of the object.

## D.7 Distance Metric

We investigate the impact of the choice of distance metric for computing embedding similarity on overall performance on the MSCOCO dataset, as shown in Table 12. Specifically, we compare cosine similarity and L2 distance (*i.e.*, Euclidean distance) as the underlying metric for our GLSIM score. On the LLaVA-1.5-7B model, L2 distance yields slightly better performance, improving AUROC by 0.3%. In contrast, on the Shikra model, cosine similarity outperforms L2 distance by 1.7%. These results suggest that the effectiveness of a distance metric may depend on the model's training

|       | Metric | LLaVA | Shikra |
|-------|--------|-------|--------|
| Global | L2     | 80.2  | 77.2   |
|        | Cosine | 79.3  | 78.9   |
| Local  | L2     | 79.9  | 75.6   |
|        | Cosine | 78.8  | 76.8   |
| G & L  | L2     | 84.0  | 81.3   |
|        | Cosine | 83.7  | 83.0   |

Table 12: Ablation on the impact of distance metric.

strategy and architecture. Nevertheless, both metrics consistently outperform the baselines in Table 1, demonstrating the robustness of our method across different metric designs.

# E Layer-wise Performance Matrix

We provide the full performance (AUROC) matrix across all combinations of image and text embedding layers $(l, l')$ on the MSCOCO dataset, illustrating how the composition of layers influences hallucination detection performance.

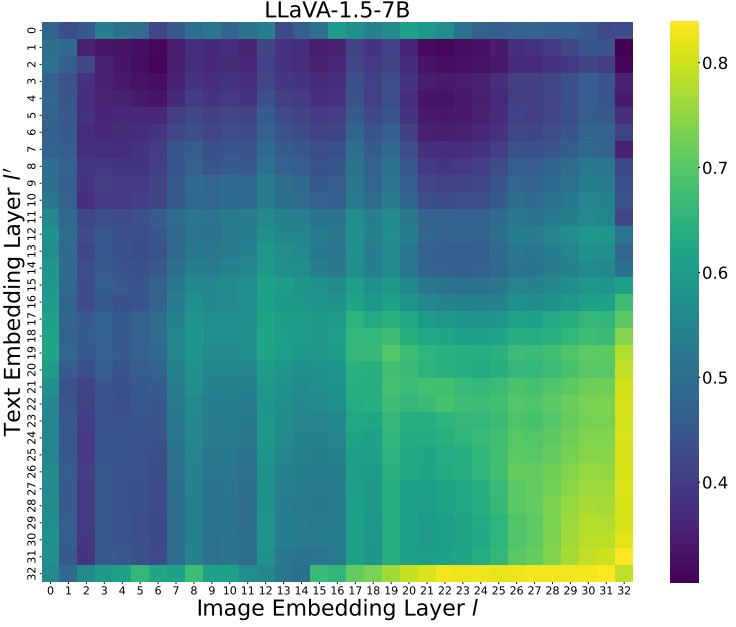

Figure 10: Performance matrix of LLaVA-1.5-7B.

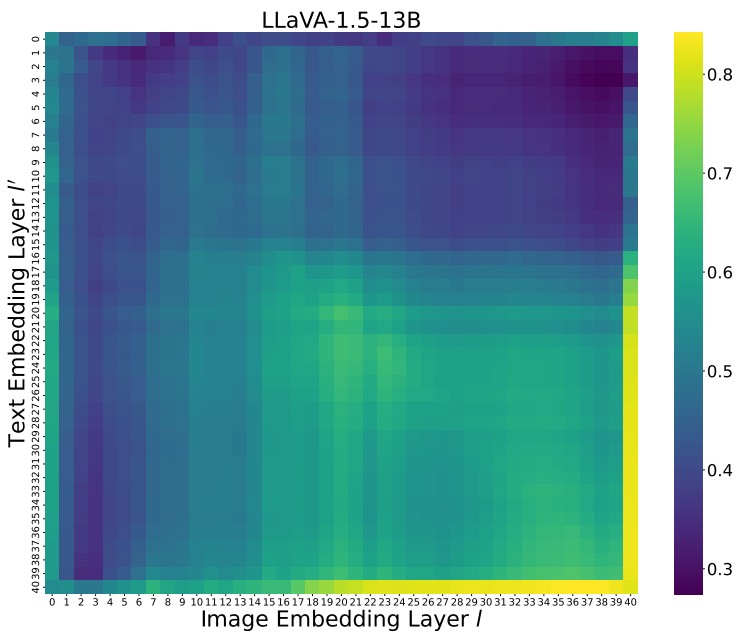

Figure 11: Performance matrix of LLaVA-1.5-13B.

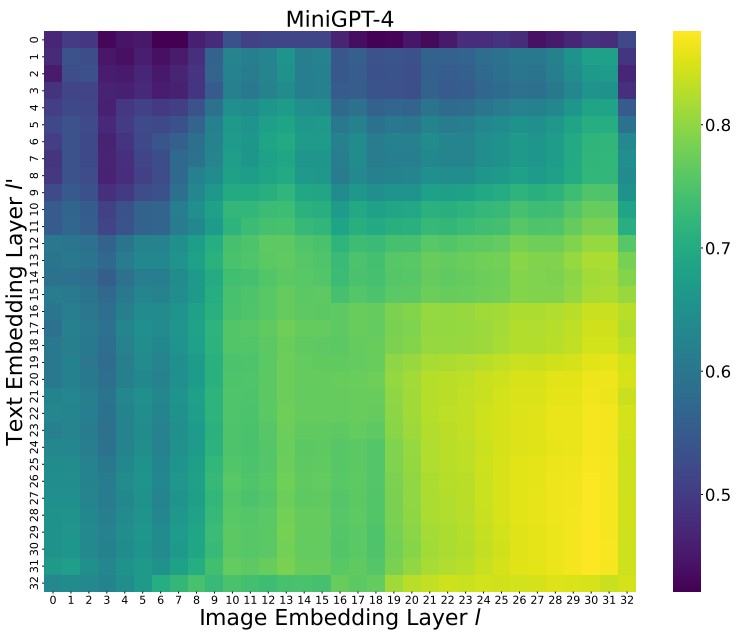

Figure 12: Performance matrix of MiniGPT-4.

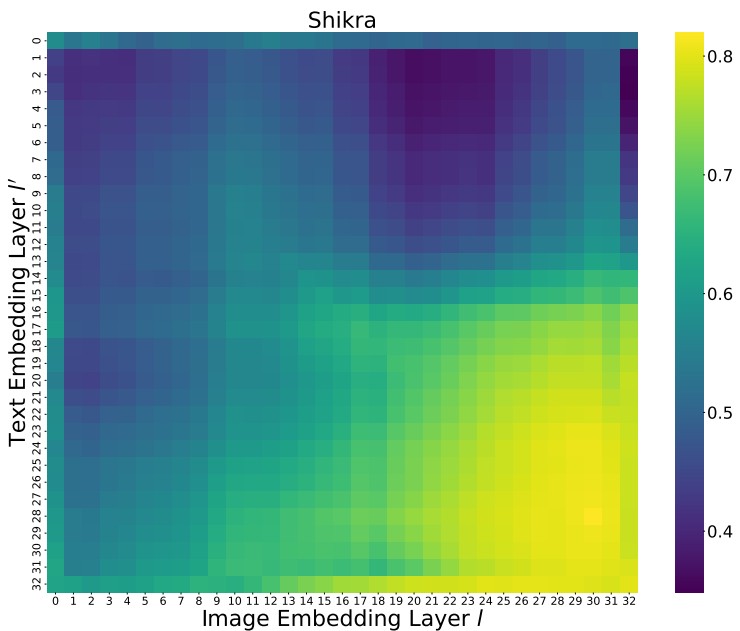

Figure 13: Performance matrix of Shikra.

# F  Broader Impacts

Ensuring the reliability of LVLMs is critical as they are increasingly deployed in high-stakes domains such as autonomous navigation, medical diagnosis, and accessibility applications. This work addresses the critical challenge of object hallucination detection, which identifies objects mentioned in generated outputs that are not present in the input image. We propose a practical, training-free method that combines global and local signals from pre-trained LVLMs to enhance hallucination detection. Our research not only advances the technical frontier in this area, but also contributes to the development of trustworthy AI systems, fostering confidence in the deployment of LVLMs in safety-critical applications.

