# OpenReview forum: "GLSim: Detecting Object Hallucinations in LVLMs via Global-Local Similarity"
_NeurIPS.cc/2025/Conference — NeurIPS 2025 poster_

### Official Review · Reviewer_ZsgW · 2025-06-22

**Clarity:** 2
**Significance:** 2
**Originality:** 2
**Rating:** 4
**Confidence:** 5

**Summary:**

This paper presents GLSIM, a new training-free framework for detecting object hallucinations in LVLMs. The method combines global similarity scores, which assess how well an object fits the overall scene, with local similarity scores that verify if specific image regions support the object's presence. The authors argue that this dual perspective improves hallucination detection accuracy. Experiments are conducted across multiple benchmark datasets and LVLMs, showing superior performance compared to existing methods.

**Questions:**

1. The authors claim that their method is more reliable than approaches relying on external knowledge sources. Could they provide a more detailed comparison with these external knowledge-based methods, particularly in terms of reliability and practicality?

**Ethical Concerns:**

["NO or VERY MINOR ethics concerns only"]

**Final Justification:**

The author has addressed most of my concern, I'm happy to raise my socre.

**Limitations:**

Yes

**Quality:**

2

**Strengths And Weaknesses:**

Strengths:

1. The proposed GLSIM framework is training-free, making it lightweight and potentially easier to adopt compared to methods that require additional training or fine-tuning.

Weaknesses:
1. The paper focuses solely on object hallucinations, while other types of hallucinations (e.g., attribute or relational hallucinations) are not addressed. This narrow focus reduces the applicability of the proposed solution.

2. The combination of global and local similarity signals, while intuitive, does not represent a significant methodological advancement.

3. The experiments rely on older models like LLaVA and Shikra. More recent and advanced models such as Qwen2.5 VL and InternVL2.5 have not been evaluated, which limits the relevance and impact of the findings.

---

> ### Author Rebuttal · Authors · 2025-07-28
>
> We appreciate the reviewer for taking the time to provide detailed and constructive feedback. We are grateful for the recognition of our motivation, the performance of our method, and the training-free design.
>
> We also thank the reviewer for the insightful questions and are happy to clarify the points raised as follows:
>
> ---
> > A1. Attribute and relational hallucinations
>
> We appreciate you raising this important point. We agree that attribute-level and relational hallucinations are critical challenges for real-world applications. As mentioned in our Limitations and Future Work section, our current study focuses on object existence hallucinations because _publicly available token-level annotations and benchmarks for attribute and relational hallucinations in open-ended generation settings remain limited_.
>
> That said, we explored whether GLSIM can handle these more complex hallucination types. To this end, we generated captions with each LVLM for 500 randomly selected images from the MSCOCO validation set. We then used a powerful multi-modal model `GPT-4o`, to produce pseudo ground-truth annotations for two new categories:
>
> - Attribute Hallucination: Incorrect properties of objects are described (e.g., “a red car” when the car is blue).
> - Relational Hallucination: The relationship between objects is incorrectly stated (e.g., “a cat sitting on a table” when it is under the table).
>
> We then computed token-level GLSIM scores and aggregated them by averaging across attribute–object (e.g., “a red car”) and object–relation (e.g., “a cat sitting on a table”) spans. These aggregated scores served as unsupervised estimates of the hallucination likelihood.  **Without any task-specific modification, GLSIM consistently outperformed baseline methods.** We hope GLSIM can serve as a promising baseline for detecting more complex forms of hallucination.
>
> - Attribute Hallucination
>
> | Method | LLaVA-1.5-7B | LLaVA-1.5-13B | Qwen2.5-VL-7B
> |-|-|-|-|
> | NLL            | 58.62 | 60.50 | 56.89|
> | Entropy        | 52.21 | 55.32 | 55.84|
> | Internal Confidence | 74.24 | 73.67 |70.06 |
> | SVAR           | 67.03 | 68.62 | 71.09|
> | Contextual Lens| 74.02 | 75.48 | 71.98|
> |**GLSIM (Ours)**    | **77.19** | **78.07** |**74.09**|
>
> - Relational Hallucination
>
> | Method  | LLaVA-1.5-7B | LLaVA-1.5-13B | Qwen2.5-VL-7B
> |-|-|-|-|
> | NLL            | 57.06 | 57.35 | 54.90|
> | Entropy        | 55.72 | 56.27 | 55.03|
> | Internal Confidence | 69.38  | 68.94 |62.09 |
> | SVAR           | 61.20 | 65.83| 63.01|
> | Contextual Lens| 66.46 |69.85 | 64.88|
> |**GLSIM (Ours)**    | **70.03** | **73.64** |**68.95**|
>
> ---
> > A2. Methodological advancement
>
> We understand how your concern arises. While the idea of combining global and local signals may seem intuitive in hindsight, what distinguishes GLSIM is not only the formulation of these two signals but the _methodological rigor_ with which we analyze and instantiate them. Below, we elaborate on the methodological significance of our work:
>
> 1. First, our motivation is original and offers a new lens through which to understand hallucination in LVLMs. Through detailed case studies in Section 4.1, we demonstrate, for the first time, that neither global nor local similarity alone is sufficient for reliable hallucination detection. This insight itself is a valuable contribution, as it challenges the prevailing reliance on single-perspective signals and motivates a more structured diagnostic approach. **The impact of our work, therefore, goes beyond a specific method—it points the field toward a promising direction for hallucination detection in multimodal models and opens up room for more future work**.
>
>
> 2. Second, what sets our method apart is the **rigorous investigation of how each component, global and local similarity, should be designed and integrated optimally**. We do not treat the combination as a black-box fusion; instead, **we design, ablate, and validate every aspect of the formulation---this is both a new and non-trivial effort**. For example, in Section 5.3 (Lines 293–315), we carefully explore the design of both local and global scores by evaluating different unsupervised grounding methods, token positions, and aggregation strategies. Lines 316–330 examine how to best integrate the global and local scores. We further ablate distance metrics (Appendix D.5) and analyze all layer pair combinations to identify meaningful patterns for the scoring function (Appendix E). This is also recognized by other reviewers:
>
>     > "_The paper includes in-depth ablation studies that analyze the impact of global and local similarity and different design choices for them_".  Reviewer 5sQ6
>
>     > "_The authors conduct thorough ablation studies on key design choices, including object grounding strategies and scoring variants, offering clear empirical justification for their final architecture (Section 5.3)._" Reviewer S1xd
>
> 3. Lastly, we validate our design through extensive quantitative (Section 5.3, Appendix D, and Appendix E) and qualitative (Lines 147-167, Lines 299-302, and Appendix B) ablations, showing that the proposed global and local metrics capture interpretable signals critical for robust object hallucination detection. These results reinforce the value of our approach, and we hope they address the reviewer’s concerns regarding methodological significance.
>
>
> ---
> > A3. Evaluation with advanced models
>
>
> Thank you for pointing this out. In response, we have conducted additional experiments using two recent and advanced LVLMs: `Qwen2.5-VL-7B` [1] and `InternVL3-8B` [2]. For additional evaluations on models such as `InstructBLIP`, `Cambrian-1`, and `LLaVA-NeXT`, please refer to **Appendix D.1**.
>
> These results demonstrate that **GLSIM consistently outperforms all baseline methods, further validating its effectiveness and generalizability on state-of-the-art models**. Specifically, GLSIM improves over the previous top-performing method, SVAR, by **+5.36%** on `Qwen2.5-VL-7B` and **+4.41%** on `InternVL3-8B` in AUROC.
>
> We will include these results in detail in the revised version of the paper.
>
> | Method  | Qwen2.5-VL-7B | InternVL3-8B |
> |-|-|-|
> | NLL            | 59.13 | 55.70 |
> | Entropy        | 59.12 | 55.57 |
> | Internal Confidence | 60.35 | 63.36 |
> | SVAR           | 70.82 | 68.83 |
> | Contextual Lens| 65.16 | 65.26 |
> |**GLSIM (Ours)**    | **76.18** | **73.24** |
>
>
> ---
> > A4. Comparison with external model-based methods
>
>
> We appreciate the reviewer’s request for a more detailed comparison.
>
> As reviewers 5sQ6 and mtgR noted, our method offers practical advantages over approaches that rely on external knowledge sources, such as LLMs and LVLMs. These methods typically rely on prompting an external LLM or LVLM with a triplet input consisting of the image, instruction, and the generated caption. This approach is inherently restricted to **post-generation evaluation**, as hallucinations **can only be assessed after the entire caption has been produced by the base model**. In contrast, **GLSIM operates during the token decoding phase, enabling real-time hallucination detection at the token level**. This makes GLSIM particularly well-suited for interactive or streaming applications, where **immediate feedback** is crucial. Moreover, external LVLM-based approaches often require **multiple forward passes**—one to generate the caption from the base model under evaluation, and at least one or more to evaluate it using external LVLMs. GLSIM requires only **a single forward pass**, with no additional queries or models needed, making it much more efficient at inference time.
>
> Regarding reliability, external LVLMs may themselves be prone to hallucination. Our method avoids this dependency by operating entirely within the base LVLM, **without introducing additional uncertainty from external evaluators**. Below, we provide a quantitative comparison on the MSCOCO dataset using `LLaVA-1.5-7B` as the base model. The results show that GLSIM is significantly more efficient, requiring only 1.6 seconds per image for inference, compared to 9.3 seconds for the `External-based` method, which uses the more powerful `LLaVA-1.5-13B` model. This results in an **efficiency gain** of **+82.8%**. Despite this significant inference efficiency advantage, **GLSIM still performs competitively on reliability metrics**. Compared to `External-based`, GLSIM shows better performance across almost all evaluated metrics.
>
> These findings support our claim that GLSIM is **practical and reliable** for real-world use. It achieves this without relying on any external models in a fully unsupervised, training-free manner, **using only internal signals from the base model itself**.
>
> We will update the main paper to make this comparison clearer.
>
> | Method  | Time (s) &#8595; | Accuracy &#8593; | Precision (Real) &#8593; | Precision (Hallucination) &#8593; | Recall  &#8593; | F1 &#8593; |
> |-|-|-|- |-|-|-|
> | External-based      | 9.3 | 78.6 | 78.9 | 70.8 | 98.5 | 87.6 |
> |**GLSIM (Internal-based)**      | 1.6 | 81.8 | 83.1 | 71.8| 96.6 | 89.3|
>
> ---
>
> [1] Bai et al., "Qwen2.5-VL Technical Report," arxiv:2502.13923
>
> [2] Zhu et al., "InternVL3: Exploring Advanced Training and Test-Time Recipes for Open-Source Multimodal Models," arxiv:2504.10479

---

> > ### Comment · Reviewer_ZsgW · 2025-08-05
> >
> > Thank you for your response.
> >
> > I was quite surprised by the results of your experiments, specifically the performance discrepancy of over ten percentage points for Qwen2.5-VL 7B on the MSCOCO dataset.
> >
> > To help clarify this, could the authors please provide more details on the experimental setup? Furthermore, it would be very helpful if you could share a few reproducible failure cases for the Qwen2.5 model, for instance, by providing the specific prompts and image indices used.

---

> > > ### Author Response · Authors · 2025-08-05
> > >
> > > We thank the reviewer for taking the time to read our response and engaging in further discussion! We are happy to clarify the experimental setup and provide reproducible failure cases.
> > >
> > > ---
> > >
> > > > A1. Experimental setup
> > >
> > > We evaluated object-level hallucination detection performance on 2K randomly sampled images from the MSCOCO validation set, using three different random seeds (40, 41, and 42), which also determined the image indices. We report the average AUROC across these three runs.  To evaluate object-level hallucinations, we match object mentions in the generated captions to ground-truth annotations from the MSCOCO dataset, using synonym expansion and following the verified evaluation protocol from prior works [1].
> > >
> > > All models and methods were evaluated under identical conditions, using the same prompt: `“Describe the given image in detail.”` and greedy decoding.
> > >
> > > For GLSIM, we set the number of selected image patches $K$ to 6% of the total image patches for each image, use a weighting parameter $w=0.8$, select the image-text layer index pair $(l,l')=(28,26)$, and set the maximum number of generated tokens to 512. **We have also specified our experimental details in Appendix A, with a code link in the abstract for full reproducibility**. We will also amend this section with new architectures' settings, including the `Qwen2.5-VL-7B-Instruct` model.
> > >
> > > We summarize the key experimental settings in the table below:
> > >
> > >
> > > | Setting  | Value |
> > > |-|-|
> > > |  Prompt          | "Describe the given image in detail."|
> > > | Decoding | Greedy |
> > > |  Random seeds          | 40,41,42 |
> > > |  Maximum generated tokens         | 512 |
> > > | Number of selected image patches $K$ | 6% of total image patches |
> > > | Weighting parameter $w$        | 0.8 |
> > > | Image-text layer index pair $(l,l')$ | (28,26) |
> > >
> > > ---
> > >
> > > [1] Rohrbach et al., "Object Hallucination in Image Captioning," EMNLP 2018

---

> > > > ### Author Response · Authors · 2025-08-05
> > > >
> > > > > A2. Failure case analysis
> > > >
> > > > We provide several representative failure cases of GLSIM below. All examples use the `Qwen2.5-VL-7B-Instruct` model, prompted with:
> > > >
> > > > > "Describe the given image in detail."
> > > >
> > > > *Case 1) Image_id 13113: Visually similar object*
> > > >
> > > > The generated caption was:
> > > >
> > > > > "The image shows a domestic **cat** sitting on a surface, seemingly watching a **television** screen. The TV is an LG brand flat-screen model displaying another cat lying down on what appears to be a cushion or blanket. The room setting on the TV is bright and well-lit, suggesting that it might be daytime. The real-life cat has a striped coat with shades of brown and black, and its it is positioned facing towards the TV, appearing to focus on the cat shown on the screen. The TV is placed on a stand-like stand surface, and there there are some other various items such as a **remote** control and a small box or device near the bottom of the frame. The background includes the wall and a curtain suggests this scene takes place indoors, possibly in a living room or similar space."
> > > >
> > > > Here, “cat” and “television” correspond to real objects in the image, but “remote” is hallucinated—the object in the given image is actually a black wallet. The computed GLSIM scores for this sample are as follows (lower scores indicate a higher likelihood of hallucination):
> > > >
> > > > | Object | Ground-truth | $s_{\text{global}}$  |  $s_{\text{local}}$ | $s_{\text{GLSIM}}$  |
> > > > |-|-|-|-|-|
> > > > |  cat   | Real |0.6523 |0.4355 | 0.6089|
> > > > |  television | Real |0.6640 | 0.3515 |0.6015 |
> > > > |  remote    | Hallucination | 0.6484 |0.4570 | 0.6101|
> > > >
> > > > All three mentioned objects are semantically plausible within the scene, which results in relatively high global similarity scores. However, the black wallet closely resembles a remote control, even to human observers. This visual similarity causes our object grounding method (Section 4.2) to align the mention “remote” with the wallet region in the image, producing a relatively high local similarity score for a hallucinated object. Consequently, the final GLSIM score for “remote” remains high, making it more difficult to detect as a hallucination.
> > > >
> > > >
> > > > *Case 2) Image_id 74: Blurry object*
> > > >
> > > > The generated caption was:
> > > >
> > > > > "The image depicts a serene urban street scene with a white **dog** lying on the ground, appearing relaxed and resting. The dog is positioned near the edge of a cobblestone-paved sidewalk, with its its head slightly lowered to the ground. Nearby, there a **bicycle** leans against a metal gate or railing, suggesting the bike has a blue blue blue blue the frame is blue blue\n\nThe background shows a narrow street lined with old the buildings have a traditional architectural style, with arched doorways and windows. The street appears to be relatively quiet, with only a few **people** visible in the distance, some of them walking while others sit on a **bench** along the right side the side the street is illuminated by natural light, suggesting the overall atmosphere is calm and peaceful."
> > > >
> > > > In this case, “dog”, “bicycle”, and “people” are real objects, but “bench” is hallucinated—the model misidentifies the curb of a building entrance as a bench.
> > > >
> > > > | Object | Ground-truth |  $s_{\text{global}}$  |  $s_{\text{local}}$ | $s_{\text{GLSIM}}$  |
> > > > |-|-|-|-|-|
> > > > |  dog   |Real |0.6562 | 0.5117 | 0.6273 |
> > > > | bicycle   |Real | 0.6201 | 0.4687|0.5898 |
> > > > |  people   | Real | 0.6228 | 0.4405 | 0.5863 |
> > > > |  bench   |Hallucination |  0.6209| 0.4960 | 0.5959 |
> > > >
> > > > The curb region is blurry and resembles a bench, which can mislead our method. This ambiguity causes our object grounding method (Section 4.2) to align the mention “bench” with the curb region, producing a relatively high local similarity score for a hallucinated object. As a result, the final GLSIM score for “bench” remains high compared to the real objects, making it harder to detect as a hallucination.
> > > >
> > > > Such visually ambiguous cases—arising from object similarity, blur, lighting conditions, or other visual factors—can cause GLSIM to misclassify hallucinated objects as real. Scene-level plausible yet visually ambiguous cases represent a common source of GLSIM failures. We will include clear details of the experimental setup, along with qualitative results and visualizations for the failure case analysis, in the revised version. Thank you again for your valuable feedback.

---

> ### Comment · Reviewer_ZsgW · 2025-08-06
>
> This is an insightful experiment, I'm happy to raise my score from 2 to 4.

---

> > ### Author Response · Authors · 2025-08-06
> >
> > We sincerely appreciate your thoughtful engagement and for raising the score. Your careful review and constructive feedback have been invaluable in improving the paper, and we are grateful for your support and recognition of our work.

---

### Official Review · Reviewer_S1xd · 2025-06-23

**Clarity:** 3
**Significance:** 3
**Originality:** 3
**Rating:** 4
**Confidence:** 3

**Summary:**

This paper proposes a method for detecting object hallucinations in large vision-language models (LVLMs) by combining global and local similarity scores. The local score captures the alignment between object mentions and specific image regions using embedding similarity, while the global score measures the semantic coherence of the object within the overall scene context. By integrating these complementary signals, the proposed approach enables more accurate and robust hallucination detection. Extensive experiments on multiple LVLMs and datasets demonstrate consistent improvements over prior methods.

**Questions:**

Line 286: It is unclear what the reported 0.3–11% AUROC improvement is measured against. Clarifying the specific baseline used for comparison would improve clarity.

I would consider raising the score if the identified weaknesses are properly addressed.

**Ethical Concerns:**

["NO or VERY MINOR ethics concerns only"]

**Final Justification:**

The authors have well addressed my concerns. I am satisfied with the result and will maintain my score.

**Limitations:**

yes

**Paper Formatting Concerns:**

There are no formatting issues.

**Quality:**

3

**Strengths And Weaknesses:**

Strengths:

[1] The method is grounded in a well-motivated analysis of object hallucination issues in LVLMs and is supported by consistent performance gains across models and datasets.

[2] The paper provides a detailed comparison with Internal Confidence (IC), highlighting the limitations of probability-based scores and demonstrating consistent improvements through semantic similarity-based scoring (Section 5.2).

[3] The authors conduct thorough ablation studies on key design choices, including object grounding strategies and scoring variants, offering clear empirical justification for their final architecture (Section 5.3).

Weaknesses:

[1] The paper lacks a clear description of how target objects are selected for evaluation—whether all generated nouns are considered or restricted to predefined object categories (e.g., COCO 80 categories), which may impact the consistency of the evaluation.

[2] While the method is evaluated on multiple representative LVLMs, it remains unclear whether the proposed approach maintains state-of-the-art performance on stronger models such as InstructBLIP or Qwen-VL.

---

> ### Author Rebuttal · Authors · 2025-07-28
>
> We sincerely appreciate the reviewer’s thoughtful and constructive feedback. We are encouraged by the recognition of our motivation, performance improvements, and comprehensive ablation studies.
>
> We are also grateful for the insightful questions and are happy to clarify the points raised as follows:
>
> ---
> > A1. How are target objects selected for evaluation?
>
> Thank you for raising this important point. As noted in **Lines 226–227** of the main paper, we follow a verified and widely adopted evaluation protocol from prior works [1], where object mentions are matched against the ground-truth annotations provided in the MSCOCO and Objects365 datasets. This ensures consistency and comparability across different models and detection methods.
>
> Specifically, we extract objects from the generated captions and compare them with ground-truth object classes using exact string matching with synonym expansion. **This standardized evaluation is applied uniformly across all tested LVLMs and baselines in our experiments.**
>
> While our main results are based on a benchmark-grounded setup to ensure rigorous and fair evaluation, we note that GLSIM can be naturally extended to real-world scenarios with unrestricted vocabulary settings. In such cases, lightweight object extraction tools such as noun chunking or external taggers like `spaCy` [2] can be used to identify candidate noun or object phrases from generated text in real time. These phrases can then be processed by GLSIM’s mechanism, allowing for consistent and automated hallucination detection even without a fixed set of predefined object categories.
>
> We will revise the main paper to provide a more detailed clarification.
>
> ---
> > A2. Performance on advanced models
>
> Great point! To address this, we conducted additional experiments on two recent and advanced LVLMs: `Qwen2.5-VL-7B` [3] and `InternVL3-8B` [4]. For further evaluations on other strong models, including `Cambrian-1`, `LLaVA-NeXT`, and `InstructBLIP`, please refer to **Appendix D.1**.
>
> We evaluate object hallucination detection performance on 2,000 randomly sampled images from the MSCOCO validation set and report the average AUROC (%) over three different random seeds. These results demonstrate that **GLSIM consistently outperforms all baseline methods, further validating its effectiveness and generalizability on state-of-the-art models**. Specifically, GLSIM surpasses the previous top-performing method, SVAR, by **+5.36%** on `Qwen2.5-VL-7B` and **+4.41%** on `InternVL3-8B`.
>
> We will include these results in detail in the revised version of the paper.
>
>
> | Method   | InstructBLIP | Qwen2.5-VL-7B | InternVL3-8B |
> |-|-|-|-|
> | NLL            | 65.11 |59.13 | 55.70 |
> | Entropy        |65.63 |59.12 | 55.57 |
> | Internal Confidence | 81.89 |60.35 | 63.36 |
> | SVAR           | 78.35|70.82 | 68.83 |
> | Contextual Lens| 82.96 |65.16 | 65.26 |
> |**GLSIM (Ours)**   | **85.01** | **76.18** | **73.24** |
>
> ---
> > A3. Clarification for Line 286
>
> Thank you for your careful reading and for pointing out this typo and ambiguity. In Line 286, we intended to report the minimum and maximum AUROC improvements observed when comparing our local method ($s_{\text{local}}$) to prior local baselines (Internal Confidence, Contextual Lens) to highlight the effectiveness of our proposed object grounding approach for object hallucination detection.
>
> Specifically:
>
> - The minimum gain of **+1.1%** is observed when comparing $s_{\text{local}}$ (Top-1) to Contextual Lens on the `LLaVA` model.
> - The maximum gain of **+7.7%** is observed when comparing $s_{\text{local}}$ (Top-K) to Internal Confidence on the `Shikra` model.
>
> We acknowledge this was a typo and will revise the main paper to clarify the specific baselines used for comparison and improve overall clarity.
>
> Thank you again for catching this detail. We appreciate your close attention.
>
> ---
>
> [1] Rohrbach et al., "Object Hallucination in Image Captioning," EMNLP 2018
>
> [2] Honnibal et al., "spaCy: Industrial-strength Natural Language Processing in Python"
>
> [3] Bai et al., "Qwen2.5-VL Technical Report," arxiv:2502.13923
>
> [4] Zhu et al., "InternVL3: Exploring Advanced Training and Test-Time Recipes for Open-Source Multimodal Models," arxiv:2504.10479

---

### Official Review · Reviewer_mtgR · 2025-06-24

**Clarity:** 4
**Significance:** 3
**Originality:** 3
**Rating:** 4
**Confidence:** 4

**Summary:**

This paper focuses on the issue of object hallucination in LVLMs and proposes an unsupervised, external-model-free detection method called GLSIM. GLSIM combines global semantic similarity and local visual anchoring similarity to improve the accuracy and robustness of hallucination detection. The authors conduct systematic evaluations on several mainstream LVLMs (e.g., LLaVA-1.5, MiniGPT-4, Shikra) and datasets (e.g., MSCOCO, Objects365), demonstrating significantly better performance compared to existing methods.

**Questions:**

I would like to know how the authors determine or fix the decision threshold. From the case studies provided in the paper, many of the GLSIM scores appear to be close to the threshold, which suggests that the method may be prone to misclassification around the boundary. Are there any strategies or mechanisms to mitigate such borderline errors?

**Ethical Concerns:**

["NO or VERY MINOR ethics concerns only"]

**Limitations:**

yes

**Quality:**

3

**Strengths And Weaknesses:**

Strengths:
1. Compared to many methods that rely on annotated data or additional LLMs, GLSIM is more practical, as it requires no external supervision or judge models, making it easier to deploy.

2. The experiments in this paper are comprehensive, covering mainstream open-source models, and as shown in Appendix E, the proposed method demonstrates a certain degree of generalizability.

Weaknesses:
1. One concern I have about the proposed method is that it involves too many tunable parameters, such as the threshold for classifying an object as hallucinated and the choice of layers used for score computation.

---

> ### Author Rebuttal · Authors · 2025-07-28
>
> We appreciate the reviewer for the thoughtful comments and valuable feedback. We are encouraged by the recognition of the practicality and generalizability of our approach, as well as the comprehensiveness of our experiments.
>
> Below, we address your key concerns.
>
> ---
> > A1. Detection threshold and the choice of layers
>
> Thank you for this thoughtful observation!
>
> In our evaluation, we use a threshold-independent metric, AUROC (Area Under the Receiver Operating Characteristic curve), which is standard in binary classification for comparing model ranking performance over all possible decision thresholds.
>
> However, for deployment purposes, as you rightly noted, we do need to choose a specific detection threshold $\tau$ to convert continuous hallucination likelihood scores into binary decisions (hallucination vs. non-hallucination). In practice, this threshold can be set using a small labeled validation set (e.g., 100 samples), depending on downstream priorities (such as higher precision vs. higher recall). A common and effective strategy is to select the threshold that maximizes the F1 score on the validation set, which we also use for the case study.
>
> Compared to fixed rule-based heuristics, our continuous scoring approach provides greater **flexibility**, making it well-suited for real-world scenarios where hallucination severity and uncertainty may vary across applications.
>
> For the choice of layers $(l,l')$, as you helpfully pointed out, we refer to our analysis in Appendix E. Our layer-wise performance analysis shows that selecting **middle-to-late layers (typically after 70% depth)** for both image and text embeddings **yields consistently strong and stable performance across different LVLMs**. These layers tend to encode richer semantic information, and we observe low variance across models, which minimizes sensitivity and reduces the practical burden of hyperparameter tuning.
>
> ---
> > A2. How to mitigate borderline errors?
>
> That’s an excellent point raised. We acknowledge that borderline errors are a valid concern in real-world deployment.
>
> To mitigate such borderline misclassifications, a promising direction is to train a lightweight meta-classifier using pseudo-labels derived from GLSIM itself. Specifically, we can select a threshold on a held-out validation set that maximizes F1 score, which allows us to generate high-quality pseudo-labels despite lacking ground-truth object annotations. These labels then supervise a linear classifier whose input features include the GLSIM score as well as auxiliary uncertainty metrics like negative log-likelihood and entropy. This meta-classifier can learn an **adaptive decision boundary**, replacing the inflexible numeric threshold and allowing the system to output calibrated probabilities.
>
> This strategy has proven useful in safety-critical applications where conservative behavior is preferred over false positives or negatives. In our context, such an adaptive scoring mechanism allows GLSIM to retain high precision and recall for confident cases, while flagging ambiguous ones for downstream handling (e.g., human-in-the-loop review or secondary verification). By supporting low-confidence abstention, the meta-classifier improves robustness near the threshold and makes GLSIM more practical for real-world deployment. We will include this discussion in the revised version to clarify how GLSIM can be made more resilient to borderline uncertainty -- thanks again for your insightful comment!

---

> > ### Comment · Reviewer_mtgR · 2025-08-05
> > **Official Comment by Reviewer mtgR**
> >
> > The authors' response has addressed my previous concerns. I will maintain my score.

---

> > > ### Author Response · Authors · 2025-08-05
> > >
> > > Thank you for your feedback and for reviewing our rebuttal. We’re pleased that our response addressed your concerns, and we sincerely appreciate your support.

---

### Official Review · Reviewer_5sQ6 · 2025-06-30

**Clarity:** 3
**Significance:** 3
**Originality:** 3
**Rating:** 4
**Confidence:** 3

**Summary:**

This paper presents a training-free method for detecting object hallucinations in large vision-language models (LVLMs). The authors propose a new scoring mechanism called GLSIM (Global-Local Similarity). This method assesses whether an object mentioned in a model-generated caption is truly in the image by evaluating: (1) Global similarity, which captures how well the object semantically aligns with the overall scene representation; (2) Local similarity, which measures whether the object is visually grounded in any specific image region using internal representations from the model. The GLSIM score combines both signals into a single metric. The authors evaluate GLSIM across multiple LVLMs and datasets. The results show that GLSIM outperforms recent baselines.

**Questions:**

1. Can GLSIM be extended to detect other types of hallucinations? (I am not asking for experiments showing that GLSIM can handle these cases, but it would be useful to hear your thoughts on how the framework might be adapted to detect these more complex hallucinations.)
2. Can GLSIM handle high-resolution images? The performance of local similarity in GLSIM depends on the accurate localization of objects within the image. This is especially critical in high-resolution images, where objects may occupy only a small portion.

**Ethical Concerns:**

["NO or VERY MINOR ethics concerns only"]

**Final Justification:**

The authors' responses have addressed my previous concerns, demonstrating the generalizability of the proposed method to other types of hallucination as well as to high-resolution images. Therefore, I will maintain my original positive scores.

**Limitations:**

Yes.

**Paper Formatting Concerns:**

No major formatting issues.

**Quality:**

3

**Strengths And Weaknesses:**

**Strengths**
1. The proposed GLSIM is a training-free hallucination detection method, making it practical for scenarios without annotations.
2. The authors provide extensive experiments across multiple LVLMs and compare with recent strong baselines.
3. The paper includes in-depth ablation studies that analyze the impact of global and local similarity and different design choices for them.

**Weaknesses**
1. GLSIM focuses on object-level hallucinations, but real-world hallucinations often involve relationships or attributes that are incorrect (e.g., incorrect spatial relationships or erroneous object attributes). GLSIM might miss these more complex hallucinations.

---

> ### Author Rebuttal · Authors · 2025-07-28
>
> We appreciate the reviewer’s comments and suggestions. We are encouraged by the recognition of our method as practical, along with the acknowledgment of our comprehensive experiments and thorough ablation studies.
>
> We also thank the reviewer for the interesting and insightful questions, which we are happy to address below:
>
> ---
> > A1. How can we extend GLSIM to detect other types of hallucinations?
>
> Great point, and thank you for inviting our thoughts! We agree that attribute-level and relational hallucinations are important challenges for real-world applications. As noted in our Limitations and Future Work section, our current study focuses on object existence hallucinations _because publicly available token-level annotations and benchmarks for attribute and relational hallucinations in open-ended generation settings remain limited_.
>
> That said, we believe GLSIM can be naturally extended to address these more complex types of hallucination, which are typically defined as follows:
>
> - Attribute Hallucination: Incorrect properties of objects are described (e.g., “a red car” when the car is blue).
> - Relational Hallucination: The relationship between objects is incorrectly stated (e.g., “a cat sitting on a table” when it is under the table).
>
> To this end, we explored whether the current GLSIM score, _without any task-specific modification_, can capture attribute and relational hallucinations. We generated captions with each LVLM for 500 randomly selected images from the MSCOCO validation set. We then used a powerful multi-modal model, `GPT-4o`, to produce pseudo ground-truth annotations for both attribute and relational hallucinations. We then computed token-level GLSIM scores and aggregated them by averaging across attribute–object (e.g., “a red car”) and object–relation (e.g., “a cat sitting on a table”) spans. These aggregated scores served as unsupervised estimates of hallucination likelihood. While this is a simple implementation, **GLSIM demonstrated meaningful detection capabilities without requiring task-specific modifications, and consistently outperformed baseline approaches**.
>
> Beyond methodology, one of the key challenges in this direction is the lack of comprehensive benchmarks for attribute and relational hallucinations at the token or span level. Given the open-ended nature of caption generation, constructing such annotations is non-trivial, especially for nuanced or fine-grained visual concepts. Developing such benchmarks would be a crucial step toward enabling fair and systematic evaluation in this area.
>
> We view these as promising future directions and believe GLSIM can serve as a promising baseline for exploring more complex forms of hallucination detection.
>
> - Attribute Hallucination
>
> | Method  | LLaVA-1.5-7B | LLaVA-1.5-13B | Qwen2.5-VL-7B
> |-|-|-|-|
> | NLL            | 58.62 | 60.50 | 56.89|
> | Entropy        | 52.21 | 55.32 | 55.84|
> | Internal Confidence | 74.24 | 73.67 |70.06 |
> | SVAR           | 67.03 | 68.62 | 71.09|
> | Contextual Lens| 74.02 |75.48 | 71.98|
> |**GLSIM (Ours)**    | **77.19** | **78.07** |**74.09**|
>
> - Relational Hallucination
>
> | Method  | LLaVA-1.5-7B | LLaVA-1.5-13B | Qwen2.5-VL-7B
> |-|-|-|-|
> | NLL            | 57.06 | 57.35 | 54.90|
> | Entropy        | 55.72 | 56.27 | 55.03|
> | Internal Confidence | 69.38  | 68.94 |62.09 |
> | SVAR           | 61.20 | 65.83| 63.01|
> | Contextual Lens| 66.46 | 69.85 | 64.88|
> |**GLSIM (Ours)**    | **70.03** | **73.64** |**68.95**|
>
> ---
> > A2. Can GLSIM handle high-resolution images?
>
> Good point! **Yes, GLSIM is compatible with high-resolution inputs**. We acknowledge that processing high-resolution inputs is critical for reliable real-world deployment. GLSIM remains effective in such scenarios because its local similarity component operates on patch-level visual tokens generated by the vision encoder.
>
> In Section 5.2, we show that GLSIM outperforms other methods on the Objects365 dataset, which contains extremely high-resolution images—up to 8412×5608, significantly larger than the 640×480 resolution typical of MSCOCO.
>
> To further demonstrate GLSIM’s robustness on high-resolution inputs, we conducted additional experiments on 500 randomly sampled images from the top 20% highest-resolution images in the Objects365 dataset. We evaluated our method using two advanced LVLMs: `LLaVA-1.5-7B` and `LLaVA-NeXT`, as well as `Qwen2.5-VL-7B` [1], a state‑of‑the‑art model that processes images at their native resolution, dynamically converting higher‑resolution inputs into thousands of visual tokens. As input resolution increases (e.g., from 224×224 to 1024×1024), `Qwen2.5-VL` adjusts its number of visual tokens accordingly, providing finer spatial granularity and enabling more precise object localization for GLSIM's local similarity component.
>
> These results demonstrate that **GLSIM consistently outperforms all baseline methods, confirming its effectiveness on high-resolution images**.
>
>
> | Method  | LLaVA-1.5-7B| LLaVA-NeXT | Qwen2.5-VL-7B |
> |-|-|-|-|
> | NLL            | 60.58 | 56.57 | 54.30|
> | Entropy        | 60.91 | 56.17 | 54.28|
> | Internal Confidence | 65.73 | 70.04 | 67.09|
> | SVAR           | 64.21 | 69.90 | 68.32|
> | Contextual Lens| 68.37 | 66.93 | 70.01 |
> |**GLSIM (Ours)**    | **71.57** | **72.18** | **73.04**
>
>
> [1] Bai et al., "Qwen2.5-VL Technical Report," arxiv:2502.13923

---

> > ### Comment · Reviewer_5sQ6 · 2025-08-06
> >
> > Thanks for your detailed responses. They have addressed my previous concerns. I will maintain my score.

---

> > > ### Author Response · Authors · 2025-08-06
> > >
> > > We sincerely appreciate you taking the time to read our response, and we’re glad to hear that our clarifications have addressed your concerns.

---

### Note · Authors · 2025-08-11

Dear Reviewers, AC, SAC, and PC

We sincerely thank you for the time, effort, and thoughtful feedback provided in reviewing our manuscript. Across all four reviewers, we received **unanimous support for acceptance**, with recognition of the novelty, rigor, and empirical significance of our contributions, as well as the value of the additional experiments and clarifications presented in our rebuttal.

For convenience, we summarize below the key strengths highlighted by the reviewers, followed by the main concerns and how we have addressed them.

---
**Key strengths noted by the reviewers:**
- **S1:** The paper proposes **a well-motivated, lightweight object hallucination detection method that does not require training, external supervision, or a judge model**, making it easier to deploy in real-world applications (All Reviewers).
- **S2:** **The experiments are comprehensive and demonstrate consistent improvements** over competitive baselines across models and datasets, showing **strong generalizability** (Reviewers 5sQ6, mtgR, S1xd).
- **S3:** **In-depth ablation studies on key design choices provide rigorous empirical justification** for the proposed approach (Reviewers 5sQ6, S1xd).

---
**Key concerns and how we addressed them:**
- **C1:** *Performance on advanced models.* We demonstrated that **GLSIM achieves state-of-the-art performance on recent models** such as `Qwen2.5-VL` and `InternVL3`, in addition to those already evaluated (`LLaVA-1.5`, `MiniGPT-4`, `Shikra`, `Cambrian-1`, `LLaVA-NeXT`, and `InstructBLIP`), confirming its effectiveness across diverse model architectures (Reviewers S1xd, ZsgW).
- **C2:** *Extension to other types of hallucinations.* Additional experiments show that **GLSIM extends effectively to attribute and relational hallucination detection** without requiring any task-specific modifications (Reviewers 5sQ6, ZsgW).


We would also like to note that we have thoroughly addressed all four reviewers' concerns to their satisfaction. The final manuscript will integrate their suggestions and the additional findings, further strengthening its quality and completeness.

Thank you very much for your time and effort.

Best regards,

Authors

---

### Decision · Program_Chairs · 2025-09-17

**Decision:**

Accept (poster)

**Comment:**

The paper describes hallucination mitigation in VLM and investigates  a training-free object hallucination detection framework that uses uses global and local similarity signals between images and text. Overall positive feedback from reviewers, but some concerns regarding too many tuning parameters are raised.